# A privacy-preserved horizontal federated learning for malignant glioma tumour detection using distributed data-silos

Shagun Sharma[1]☯, Kalpna Guleria[1]☯, Ayush Dogra[1]☯, Deepali Gupta[1]☯, Sapna Juneja[2], Swati Kumari[3]*, Ali Nauman[4]☯

1 Chitkara University Institute of Engineering and Technology, Chitkara University, Rajpura, Punjab, India, 2 KIET Group of Institutions, Ghaziabad, India, 3 Trinity College Dublin, Dublin, Ireland, 4 Department of Computer Science and Engineering, Ycungnam University, Gyeongsan, South Korea

☯ These authors contributed equally to this work.
* kumaris@tcd.ie

**Data Availability Statement:** All dataset files are available from the below mentioned URL: https://www.kaggle.com/masoudnickparvar/brain-tumor-mri-dataset.

## Abstract

Malignant glioma is the uncontrollable growth of cells in the spinal cord and brain that look similar to the normal glial cells. The most essential part of the nervous system is glial cells, which support the brain's functioning prominently. However, with the evolution of glioma, tumours form that invade healthy tissues in the brain, leading to neurological impairment, seizures, hormonal dysregulation, and venous thromboembolism. Medical tests, including medical resonance imaging (MRI), computed tomography (CT) scans, biopsy, and electro-encephalograms are used for early detection of glioma. However, these tests are expensive and may cause irritation and allergic reactions due to ionizing radiation. The deep learning models are highly optimal for disease prediction, however, the challenge associated with it is the requirement for substantial memory and storage to amalgamate the patient's information at a centralized location. Additionally, it also has patient data-privacy concerns leading to anonymous information generalization, regulatory compliance issues, and data leakage challenges. Therefore, in the proposed work, a distributed and privacy-preserved horizontal federated learning-based malignant glioma disease detection model has been developed by employing 5 and 10 different clients' architectures in independent and identically distributed (IID) and non-IID distributions. Initially, for developing this model, the collection of the MRI scans of non-tumour and glioma tumours has been done, which are further pre-processed by performing data balancing and image resizing. The configuration and development of the pre-trained MobileNetV2 base model have been performed, which is then applied to the federated learning(FL) framework. The configurations of this model have been kept as 0.001, Adam, 32, 10, 10, FedAVG, and 10 for learning rate, optimizer, batch size, local epochs, global epochs, aggregation, and rounds, respectively. The proposed model has provided the most prominent accuracy with 5 clients' architecture as 99.76% and 99.71% for IID and non-IID distributions, respectively. These outcomes demonstrate that the model is highly optimized and generalizes the improved outcomes when compared to the state-of-the-art models.

**Funding:** The author(s) received no specific funding for this work.

## 1. Introduction

The human body is an intricate and complex system that consists of various tissues, cells, the immune system, and organs [1]. The tissues are the group of similar cells that perform specific functions to maintain the balance between human organs [2]. It also maintains harmony and acts on the body changes by repairing damages and adapting to the new environment. Each day, a number of cells are created through the cell cycle process, which has three phases, including interface, mitosis, and cytokinesis corresponding to cell division, copying DNA and cytoplasm, and creating two different cells [3]. The cell cycle process is common in each human body, however, if the cells replicate faster or have uncontrollable growth, it can cause the origin of tumours. It produces multiple tissues in the body, which can either be benign or malignant, potentially leading to mental disability, obstruction, infections, and even death. The abnormal growth of the cells can affect the entire human body and disturb its complete functioning, leading to fatal outcomes [4]. The occurrence of tumours may range from non-cancerous (benign) to cancerous (malignant), and the treatment of malignant tumours is challenging due to their rapid growth. As per the report published by WHO [5], cancer is the second most common cause of increased mortality globally. In 2018, 9.6 million deaths have been reported equating to at least 1 death caused by cancer among 6 deaths. There are different types of cancers, including prostate, breast, colorectum, lung, and bone cancers causing impaired mobility, metastasis, immune system suppression, and psychological impacts. Glioma is a similar kind of cancer, which originates in the central nervous system (CNS), impacting neurological functions and cognitive abilities and causing seizures [6, 7]. In the report published by the National Library of Medicine [6, 8], it has been mentioned that in the years 1995–2018, a number of 31,922 deaths were reported caused by glioma tumours. It is one of the deadliest diseases if detected in advanced stages [9]. In the last few years, the number of deaths caused by malignant tumours has rapidly increased, which led to healthcare system strain, public health concerns, and societal and psychological impacts [6, 8, 10]. Therefore, there is a need to diagnose the presence of glioma early to result in effective treatment. In the healthcare departments, basic tests, including MRI, CT scans, PET scans, biopsy, and electroencephalograms, are used to detect glioma. However, MRI and CT scans are expensive and may cause irritation and allergic reactions due to ionizing radiation, and biopsy can cause pain and may outcome in swelling in the incision site. The screening and accurate detection of glioma requires expertise and specialization [4, 11]. However, in Indian remote areas, the availability of doctors and health practitioners is less, which results in advancing the stages of such diseases, causing an increased fatality rate. Hence, in numerous studies, deep learning (DL) techniques have been implemented for the early detection of malignant glioma [6, 8, 10, 12]. The convolutional neural networks (CNN), VGG16, InceptionV3, DenseNet, and U-Net were used in [13–16] for the early identification of brain tumours. These models are privacy-sensitive and work in the centralized framework for achieving accurate prediction outcomes. The Health Insurance Portability and Accountability Act. (HIPAA) has been enacted in 1996 and is followed by all the healthcare organizations in the United States to maintain and secure the health information of patients. Hence, data collection becomes the biggest challenge leading to difficulty in implementing the DL due to the requirement of a vast amount of information to provide generalized and accurate predictions. Another challenge associated with the DL models is the requirement for substantial memory and storage to amalgamate the patient's information at a centralized location. Therefore, federated learning was introduced by McMahan et al. in 2016 for privacy-protected disease detection model development, [17–19]. This architecture has several advantages over the traditional centralized framework, including data privacy preservation, less computation overhead, improved model communication efficiency,

and the ability to deal with intricate and diverse datasets [20, 21]. In this work, a federated learning-based (FL-based) model has been proposed for the early prediction of malignant glioma tumours. The base model for the development of the proposed framework has been kept as pre-trained MobileNetV2. Initially, the collected dataset was balanced to reduce the class imbalance, effectively train the model, and add uniformity in data representation. Further, the pre-processed images with the resized images to 224X224 have been fed to the model to operate within a federated architecture comprising 5 and 10 clients in independent and identically distributed (IID) and non-IID distributions. The IID distribution follows the principle of dividing the complete dataset into a number of shards to the clients, where the data contained by all the clients are different, and for non-IID, the data distribution remains different along with the dissimilarity among the information contained within clients.

The key contributions of the proposed work have been mentioned below:

- This work proposes a privacy-protected federated learning-based glioma detection model that operates within a collaborative architecture and ensures compliance with healthcare data protection regulations, namely, the General Data Protection Regulation, the Personal Information Protection and Electronic Documents Act., and the Digital Personal Data Protection Act., etc.

- The proposed glioma detection federated learning model was applied in IID and non-IID distributions with an intricate and diverse dataset containing glioma and no tumour magnetic resonance images.

- The model employed a FedAVG aggregation technique with the Adam optimization on 5 and 10 client federated architectures. This configuration implementation aimed at achieving the most prominent and enhanced converged outcomes for malignant glioma detection in sparse datasets.

The proposed article has been divided into multiple sections. Section 2 provides a summary of the techniques, datasets, hyperparameter configuration, and research scopes of the existing tumour detection models. Section 3 discusses the methodology for the proposed federated learning-based malignant glioma tumour detection model. Section 4 elaborates on the results obtained by the proposed federated learning-based MobileNetV2 model, and section 5 provides the conclusion of the work.

## 2. Literature review on tumour detection federated learning frameworks

This section discusses the literature review on tumour detection using DL and FL models.

### 2.1 Literature review on tumour detection deep learning frameworks

Various researchers are working on the DL models for glioma detection. These models have been found as the most proficient in detecting diseases at the early stages. This section discusses the DL models for early glioma detection in MRI scans.

In [22], the authors introduced a probabilistic neural network (PNN) for the prediction of glioblastoma in MRI scans. The MRI dataset is curated from Github, and pre-processing is performed to resize the images and perform histogram equalization. The results of the model depict that the model is highly efficient in predicting the accurate class with a value of accuracy as 90.9%. However, the model needs to improve in terms of protecting the privacy of the patient's data due to its collection at the centralized server. In [23], an ensemble model for the classification of glioma tumour using CNN as the feature extractor and support vector

machine (SVM) as the classifier. The classification of HGG and LGG has been done with an accuracy of 96.19% and 95.46%, respectively.

In [24], the authors have introduced a brain tumour segmentation model using ML methods. The preprocessing of the images has been performed using the histogram equalization method. Further, the segmentation is done on the pre-processed images using clustering-based methods, and optimization techniques have been applied. In [25], the authors introduced a CNN model for classifying brain tumours using MRI images. The dataset used for the implementation comprised 3264 images, which have been pre-processed and classified with an accuracy of 93.3%. The work presented in [26] tumour detection using VGG19, ResNet152, MobileNetV3, and DenseNet169 with 7023 MRI images. The highest accuracy has been achieved by the MobileNetV3 model as 98.52%. In addition, the highest training time and lowest training time have been taken by MobileNetV3 and DenseNet169 models, respectively.

## 2.2 Literature review on tumour detection federated learning frameworks

In [27], authors proposed a FL framework for brain tumour prediction using two datasets, namely, BT-large-3c and BT-small-2c containing 3,264 and 253 images, respectively. The pre-processing of the datasets has been performed by utilising image resizing, cropping, and thresholding operations. The basic CNN architecture was applied to the FL framework, where the input image size of 224X224 was used. Further, the accuracy performance has resulted in 82% and 86% for BT-large-2c and BT-small-3c datasets, respectively. The work presented in [28] the FL framework for brain tumour classification using MRI images. The authors have implemented six different pre-trained models for training, out of which three best-performing models, namely, InceptionV3, DenseNet, and VGG16, have been found and combined to form an ensemble model and implemented on the FL framework. The model implemented in DL and FL has shown an accuracy of 96.68% and 91.05%, respectively.

In [29], the development of a FL-centric approach for the detection of breast cancer has been done. In this model, the images have been processed through two different feature extractors, ResNet and GaborNet. Initially, the input images were substituted for the ResNet model, and the extracted features were further provided to the GaborNet model, which was finally substituted for a custom classifier for detecting breast cancer. The implementation of this model has been done using two different datasets containing histopathological images with 2,77,524 and 9,109 images. The results of the proposed FL-centric approach were compared to the centralized DL techniques, which shows that the proposed technique outperforms by showing better specificity. In [30], the authors implemented the pre-trained VGG model in a FL-centric environment for the identification of tumour-infiltrating lymphocytes. The dataset collection has been done from 12 different anatomical sites that contained the cancer images. The performance outcomes of the technique have resulted in an accuracy of 89% as an average of the complete accuracies identified at 8 different clients.

The model presented in [31] has been implemented using curriculum learning in an FL environment for breast cancer identification. For the model development, the authors curated 3 different datasets namely, Hologic, GE, and Siemens containing 1460, 410, and 852 images, respectively. The performance of this model has been analyzed in terms of ROC-AUC and PR-AUC. In [32], the authors employed a breast cancer detection model using FL-based pre-trained models, including ResNet50, DenseNet201, MobileNetV2100, and EfficientNetB7. The data has been collected from the BreakHis website having the breast cancer images of benign and malignant as 2480 and 5429, respectively. The performance comparison of each model has resulted in the ResNet50 model outperforms by showing a 5.7% improvement in comparison to the MobileNetV2100 model. In [33], a FL-based Cox model for predicting survival rate

from larynx tumours has been proposed. The collected dataset contained 1821 attributes which were utilized to train the Cox model. The base model for the FL framework developed was used as regression, and the dataset was provided to 3 clients. The authors in [34] proposed a FL framework for classifying infiltrating lymphocytes. The dataset was collected from 12 different sites, which were further used to train the model using the FL-based VGG16 model. The implementation of the model was performed with the resized images to 300X300 pixels, with the hyperparameters tuning of activation function, optimizer, and learning rate at sigmoid, Adam, and 0.001, respectively.

The authors of [35], have developed a FL-centric model which is based on U-Net architecture. This model is implemented for brain tumour segmentation, where the dataset, namely the "BRATS dataset" containing 4 classes of brain tumour MRI images, has been collected. The collected MRIs have been pre-processed by performing the Z-score normalization technique. This FL-based architecture has been implemented with 5 local training epochs, which resulted in the highest sensitivity and specificity of 0.9 and 0.95, respectively. In [36], authors developed a FL-centric model for breast cancer classification, where the ensemble of CNN and linear model has been used for feature extraction and cancer classification. The dataset used for the model training was imbalanced which was further applied with the Synthetic Minority Over-sampling Technique (SMOTE) techniques, and the balanced dataset was utilized for model training. The authors applied the FL-centric approach in 4 clients' architecture for 60 communication rounds, and the results have identified that the proposed technique performs better in contrast to existing techniques.

Table 1 incorporates the summary of the existing cancer detection models implemented using DL-based and FL frameworks and discusses the techniques, datasets, hyperparameter configuration, and research scope. In the existing DL-based and FL-based models, it has been analyzed that various authors have applied data balancing and pre-processing techniques for implementing glioma detection models. In [27], the authors used a histogram equalization pre-processing technique, in which the contrast of the images has been enhanced. This technique adjusts the intensity distribution of the images, redistributes the values of the pixels, and results in increased brightness levels, leading to better and enhanced contrast in comparison to the original image. This technique is used for grayscale images to enhance the quality by representing the frequency of each pixel in the images. Further, in [29], the authors used an improved histogram equalization pre-processing technique called contrast limited adaptive histogram equalization (CLAHE), which enhances the local contrast of the images, unlike globally operating on each pixel. In the CLAHE technique, the small regions of the images have been applied with equalization and contrast enhancement. It resulted in images with better and enhanced contrast at each level without creating noise in the contrast-enhanced regions. In [28], the authors used skull stripping and co-registration pre-processing techniques, which are generally used for medical imaging. The technique is used to remove the scalp, skull, and external cerebral structures from the images. This process eliminates unnecessary structures, improves detection accuracy, and standardizes the image dataset. In [25], the authors used a Laplacian and Gaussian filtering pre-processing technique for image enhancement by detecting edges and reducing noise. In Gaussian filtering, the authors used a Gaussian function to create a convolutional kernel for assigning weight values to each pixel. It simply assigns the weight based on the distance between the pixel and the center of the kernel. Further, the Laplacian filter was used to highlight the pixels with the instant intensity change. This filter has been used to detect edges in the images and improve accuracy by suppressing the slow-intensity pixels and highlighting the pixels with high spatial resolutions. In [27], the authors used thresholding, dilation, gaussian bluer, erosion, outer contouring, and extreme pointing techniques for image pre-processing. In thresholding, the pixels with the higher

**Table 1. A summary of the technique, dataset, hyperparameter configuration, and research scopes of the existing tumour detection models implemented in the FL framework.**

| Ref. | Technique | Dataset | Dataset Balancing | Hyper-parameters | Pre-processing | Accuracy | Future Scope |
|---|---|---|---|---|---|---|---|
| **Deep Learning Frameworks** | | | | | | | |
| [22] | PNN | 150 images | ✖ | NM | ✓ **Method**: Histogram Equalization for increasing the lower contrast to the upper contrast. | 90.9% | The model is highly privacy-sensitive and requires the implementation of a security mechanism to develop a privacy-protected model. |
| [23] | CNN+SVM | BraTS dataset | ✖ | NM | ✓ **Method:** Skull Stripping and co-registration | HGG: 96.19% LGG: 95.46% | Limited information on the hyperparameters has been provided, which may produce challenges in model reproducibility. |
| [24] | Clustering model | MRI images | ✖ | NM | ✓ **Method:** Contrast Limited Adaptive Histogram Equalization (CLAHE) | NM | The authors have applied the clustering models for segmentation. However, various DL models, such as UNet, are already available for image segmentation. However, the authors have not applied any DL models for segmentation purposes. |
| [25] | CNN | MRIC image: 3264 images | ✖ | Activation: ReLU, Learning rate: 0.01, Epoch: 80, Batch size: 18 | ✓ **Method:** Laplacian and Gaussian filtering | 93.3% | The authors have applied the CNN model, however, the model is highly privacy sensitive due to a centralized framework. |
| [26] | VGG19, ResNet152, MobileNetV3 and DenseNet169 | 7022 images | ✖ | Activation: ReLU, Learning rate: 0.001, Epoch: 50 | ✓ **Method:** Not mentioned | MobileNetV2: 98.52% | The number of classes are four in the proposed work, however, the dataset is small and the balancing is also not performed. |
| **Federated Learning Frameworks** | | | | | | | |
| [27] | CNN | BT-large-3c: 3264 images, BT-large-2c: 253 images | ✖ | Client: 2 Batch Size: 32 Client utilisation: 100% Learning rate: 0.001 Epoch: 45 Rounds: NM Optimizer: Adam | ✓ **Method:** Thresholding, dilation, gaussian bluer, erosion, outer contouring, extreme pointing | BT-large-3c: 86%, and BT-large-2c: 82% | The data balancing has not been performed, which may result in overfitting to the majority class, convergence issues, and poor model generalization. |
| [28] | Combined InceptionV3, DenseNet, VGG16 | 2309 images | ✖ | Client: 50 Batch Size: 16 Client utilisation: 100% Learning rate: 0.001 Epoch: 10 Rounds: 5 Optimizer: Adam | ✓ **Method:** Converting NIfTI format images to PNG | 91.05% | The dataset utilized by the authors has been identified as small compared to the number of clients taken for the training. This may result in overfitting challenges. |

(*Continued*)

**Table 1.** (Continued)

| Ref. | Technique | Dataset | Dataset Balancing | Hyper-parameters | Pre-processing | Accuracy | Future Scope |
|---|---|---|---|---|---|---|---|
| [29] | Ensemble of ResNet and GaborNet as feature extractor and custom classifier | Dataset1: 2,77,524 and Dataset2: 9,109 image | ✖ | Client: 3 Batch Size: 512 Client utilisation: 100% Epoch: 2 Learning rate: 1.0 Averaging technique: FedAVG Rounds: 30 Optimizer: SGD | ✓ **Method:** z-score normalization | Accuracy on Client 1: 84.02%, Accuracy on Client 2: 82.84%, Accuracy on Client 3: 84.88%, Overall accuracy: 83.91% | The authors have trained the model with a large batch size value, which may result in memory constraints, optimization problems, and diminishing outcomes. |
| [30] | FL-VGG | 8 datasets were collected from 12 different WSIs | ✖ | Client: 8 Epoch: 1 Learning rate: 0.001 Rounds: 500 Optimizer: Adam | ✖ | Accuracy: 89% | The configuration of the hyperparameters has not been completely discussed and mentioned in the work, which may result in difficulty for other researchers to understand the workings of this model. |
| [31] | Curriculum learning with FL | 3 datasets from GE, Hologic, and Siemens vendors containing 852, 1460, and 410 images, respectively | ✓ **Method:** Adaptive weighting | Epoch: 50 Learning rate: 0.00001 Averaging technique: FedAVG Optimizer: Adam | ✓ **Method:** Standard normalization | AUC: 0.84 PR-AUC: 0.85 | The details on the hyperparameters configuration have not been mentioned and the dataset utilized for the model implementation is small resulting in overfitting results in the class having a higher number of instances. |
| [32] | FL-based ResNet50, DenseNet201, MobileNetV2100, and EfficientNetB7 models | 7909 images | ✖ | Client: 11 Batch Size: 32 Client utilisation: 100% Learning rate: 0.001 Epoch: 5 Rounds: 20 | ✖ | NM | The dataset utilized by the authors is small when compared with its distribution among the clients. Each client has been provided with fewer images, which may result in poor model generalization and inoptimal performance outcomes. |
| [33] | Regression method | 1821 attributes | ✖ | Client: 3 | ✖ | NM | The authors have provided limited information on the hyperparameters, which may result in replicability issues and misinterpretation of the model capabilities. |
| [34] | FL-based VGG16 | NM | ✖ | Client: 8 Client utilisation: 100% Learning rate: 0.001 Epoch: 1 Rounds: 500 | ✓ **Method:** Not mentioned | Centralized model: 75% FL-model: 89% | The challenge associated with this work is that the model has a higher time complexity due to a large value of round configuration, yet, the model is not capable of achieving optimal accuracy. |

(*Continued*)

**Table 1.** (Continued)

| Ref. | Technique | Dataset | Dataset Balancing | Hyper-parameters | Pre-processing | Accuracy | Future Scope |
|---|---|---|---|---|---|---|---|
| [35] | U-Net | 285 patients MRI images | ✖ | Client: 50–100 Batch Size: 16 Client utilisation: 100% Aggregation: FedAVG Learning rate: 0.001 Epoch: 5 | ✔ **Method:** Skull stripping, z-score normalization, and image registration | Sensitivity: 0.9, Specificity: 0.95 | Though the authors applied the FL-centric approach for enhancing data security, the computational overhead is the major concern identified in this work. In addition, the small dataset has been divided into a large number of clients, which is highly unscalable if done in real-life scenarios. |
| [36] | Ensemble of CNN and linear classifier | 2620 images | ✔ **Method:** SMOTE (synthetic minority oversampling technique) | Client: 4 Batch Size: 32 Client utilisation: 100% Learning rate: NM Epoch: 3 Averaging technique: FedAVG Rounds: 60 Optimizer: SGD | ✔ **Method:** Oversampling, decision threshold, undersampling, class weight | NM | The authors implemented the model for a small number of epochs which may result in suboptimal performance due to less training, hence, the model results can be improved by increasing the value of training epochs. |

threshold were changed to white, and the remaining pixels were kept as black. In dilation and erosion, the gaps between the objects in the images have been connected, and the Gaussian blur was used to soften the images by reducing noise. Outer contouring was utilized to structure and highlight the shape of the objects in the images. Further, in extreme pointing, the orientation and boundaries of the tumour have been defined. In [29], the authors used a Z-score normalization technique in which the rescaling was performed, where the mean has been set to 0 and the standard deviation is set to 1. In [36], the authors used oversampling, decision threshold, undersampling, and class weight, where the oversampling was applied to increase the images in the minority class, and undersampling was used to decrease the images in the majority class. Further, the decision threshold has been used for classifying tumour images from non-tumour classes by setting the minimum threshold value as 0.5. Furthermore, the adaptive weighting and SMOTE (synthetic minority oversampling technique) data balancing were performed in [31, 36], respectively. Adaptive balancing was performed by dynamically adjusting the weights of tumour and non-tumour classes during the training phase, and SMOTE was performed to balance the dataset by increasing the samples in the minority class.

Though the authors applied various pre-processing and data-balancing techniques, however, these methods are limited and don't provide the solution to handle temporal dependencies in the dataset. To develop a robust glioma detection model, a broader set of pre-processing methods is required to handle the complexity of the real-world dataset.

The literature review identifies various researchers working on FL for tumour detection. The review demonstrates the effectiveness of FL in the healthcare sector while maintaining data privacy. However, various research gaps, including high computational overhead, data imbalance, and dividing of the limited dataset to numerous clients, have been analyzed in the existing studies [32, 35]. The data distribution among clients is required to be high enough to train the base model, whereas, in the studies, a small dataset has been used to implement the

FL framework [28, 36]. This limits the model to optimally training itself and results in poor model formation and result generalization. In addition, the summary of the hyperparameters configuration is not mentioned in [30–33], which results in difficulty in understanding the complete architecture of the model. Further, the limited dataset distributed among clients also results in overfitting challenges to the class, which has a higher number of instances. Though the FL may improve the performance for real-world applications in healthcare, to achieve precise disease image classification in hypothetical scenarios, a substantial dataset is essential to effectively train the base models.

The proposed model is an enhanced and improved framework, which has been trained using two different optimizers, namely, Adam and SGD, on the 5 and 10 clients with both IID and non-IID distributions. However, the models provided in the existing studies are either configured to Adam/SGD optimizer, or the client values are also kept to a particular value. There is no performance comparison with different optimizers has been made, and a varying number of clients have been formed. The performance of the proposed model has been found as optimal, which shows that with the varying clients and optimizers in IID and non-IID distributions, the model performs better in comparison to the state-of-the-art FL-based techniques.

## 3. Methodology

This section discusses the dataset and methodology of the proposed FL-based MobileNetV2 model developed for glioma tumour detection in MRI images.

### 3.1 Dataset

The tumour detection dataset has been curated from [37], which contains four classes, namely glioma, meningioma, pituitary, and non-tumour having 7022 images. There are higher similarities between non-tumour and glioma MRIs due to diffuse borders rather than sharp edges, and glioma is one of the most fatal brain tumours among meningioma and pituitary. This poses a challenge to radiologists in accurately identifying the tumour visibility. Therefore, to classify non-tumour and glioma, the valid images belonging to the corresponding classes have been extracted for the model development.

Table 2 tabulates the distribution of the glioma dataset, where the total images in the glioma and no-tumor dataset contained 2642 images.

**3.1.1 Image pre-processing.** The extracted dataset contained 2,916 training and 705 testing images, which have a class imbalance. To avoid the overfitting caused by class imbalance, each folder belonging to training has been provided with the equal number of images in each class and the ratio of training and testing was kept as 80:20. This step resulted in 3,242 images with 2594 MRIs for training and 648 MRIs for testing purpose. Further, the images also had size variations. Therefore, each image was resized to 224X224 pixels to substitute as input to the FL-based MobileNetV2 model. An example of the initially collected images and the resized images has been illustrated in Fig 1. The dataset has 2 classes, where the example of resizing the images have been shown as; sample 1 and sample 2 represent the resizing of actual no-

**Table 2. Distribution of the glioma dataset.**

| Dataset Split | Number of Samples | Percentage of Total | Purpose |
|---|---|---|---|
| Train Set | 2114 | 80% of 2642 images | Training of the model with the labeled dataset |
| Validation Set | 528 | 20% of 2642 images | Utilized for hyperparameter tuning and model selection |
| Test Set | 480 | 80% of 600 images | Used for the evaluation of the trained model |
| Total Dataset | 2642 | 100% | Complete training and testing dataset |

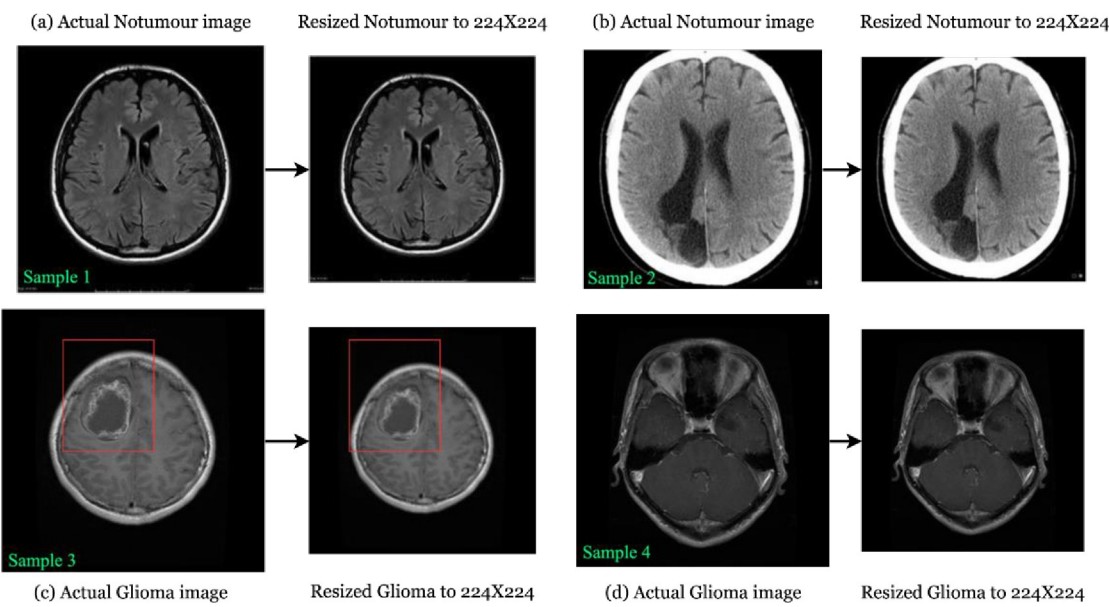

**Fig 1. Pre-processed non-tumour and glioma images.**

tumour image to 224X224 pixels and sample 3 and sample 4 shows the resizing of actual glioma image to 224X224 pixels.

## 3.2 Proposed methodology

Data privacy is one of the major concerns of the DL models due to the process of collecting the patient's information in a centralized framework. The DL models generalize optimal outcomes with the large volume of datasets. Data scarcity can hinder the capability of these models and limit their ability to identify significant outcomes. Therefore, collaborative learning processes are utilized by various researchers to collect the patients' information through a shared pool. Though this process allows the development of optimal models for disease detection, it raises the challenge of data privacy.

The proposed work introduces a FL-based MobileNetV2 model for brain tumour glioma classification from non-tumour MRI scans. The base MobileNetV2 model has been initially trained with the ImageNet dataset on the central server and distributed to the participating client in the FL framework. Each layer in the MobileNetV2 has been initialized with the pre-trained weights of the ImageNet dataset, which contains 1 million images and 1000 different classes. The MobileNetV2 model training using the ImageNet dataset is responsible for configuring the initial weights to the server and participating clients. The range of these weights is between min = -1.061939001083374 to max = 1.4804697036743164. These weights obtained from the ImageNet dataset provide better model training and faster convergence. In the proposed FL-based model, the initial weight values assigned to each client remains the same as the global model. These weight values to each client typically come from the central server, where global model initialization has been performed using a pre-trained MobileNetV2 model trained with the ImageNet dataset. In the proposed FL-model, the initial weights distributed to each client are the same as the global MobileNetV2 model trained using ImageNet dataset. It represents that each client receives and starts training with the same weights of the initial global model developed on the server side. These weight values provide the model with information regarding the classification task and lead to optimal outcomes as the model is already

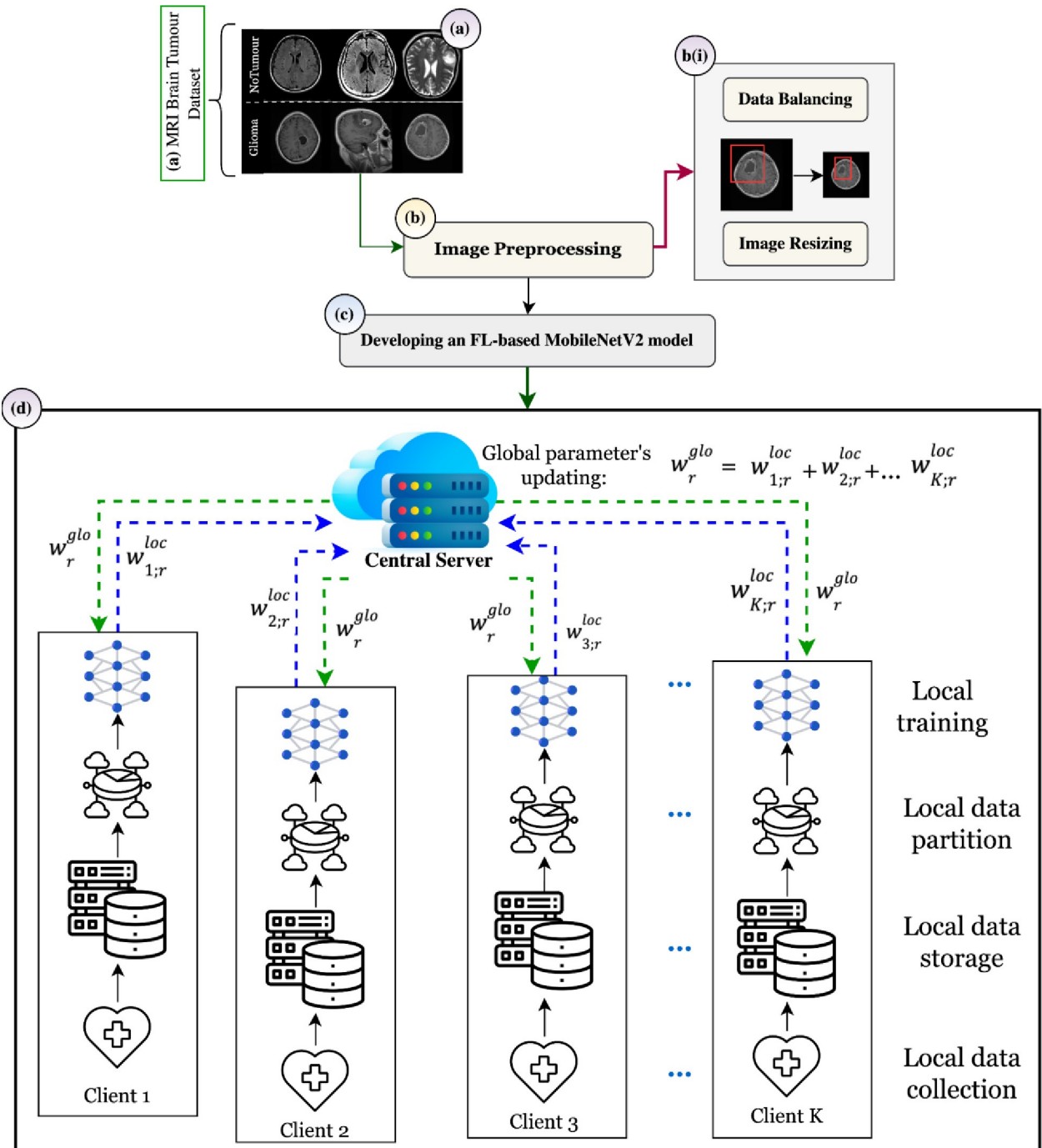

**Fig 2. Glioma brain tumour detection using privacy-protected FL framework.**

trained using a large dataset. Further, this pre-trained base model extracts the essential features from the pre-processed images.

The proposed privacy-preserved federated framework has been shown in Fig 2, illustrating four different components:

(a) The collection of the MRI scans of non-tumour and glioma tumours.

(b) The pre-processing of the collected images by performing data balancing and image resizing.

(c) Configuring and developing the pre-trained MobileNetV2 base model trained using ImageNet dataset at the server side.

(d) Distributing pre-trained MobileNetV2 to each client in the FL framework. This step includes the training of pre-trained MobileNetV2 model with the local datasets of each client. In the initial phase, each client receive a $w_r^{glob}$ model from the server, which is further trained with the local datasets $L_1$, $L_2$, $L_3$, ...., $L_K$. After the successful training, the local model updates are sent to the server as $w_{1;r}^{loc}$, $w_{2;r}^{loc}$, $w_{3;r}^{loc}$, ..., $w_{K;r}^{loc}$. Further, the local updates are collected by the server and aggregation using FedAVG technique has been performed as provided below:

$$\text{Aggregation} = w_{1;r}^{loc} + w_{2;r}^{loc} + w_{3;r}^{loc} + \ldots + w_{K;r}^{loc}$$

This aggregated value is used to update the global model and which is again sent iteratively to each client until the optimal model is developed. This framework is decentralized and preserves healthcare data privacy by aggregating the updates instead of collecting the healthcare data at the central location. In practical scenarios, this framework contains numerous hospitals and a central server, where each hospital contains its own private dataset at a personalized location. The data contained within each client is neither shared with the central server nor with the other participating clients. The federated process starts from the central server containing the base model, which is distributed to each participating client to train using a personalized dataset. The clients use their private dataset for training the base model and share the model updates with the central server, which then performs the local parameter aggregation and updates the global model with the aggregated data. Further, this process continues until the federated framework achieves the most optimal accuracy for glioma detection in a privacy-preserving architecture. Healthcare providers follow data privacy preservation acts, including HIPAA in the United States, the General Data Protection Regulation (GDPR) in the European Union, the Personal Information Protection and Electronic Documents Act (PIPEDA) in Canada, and the Digital Personal Data Protection Act. (DPDT 2023) in India [38, 39]. The proposed federated framework also follows privacy preservation making it technically suitable for disease prediction, without sharing the data to the server or any other peer clients.

The proposed federated architecture has been implemented on Google Colab with the NVIDIA T4 GPU virtual machine using Tensorflow framework version 2.17.0 integrated with TensorFlow.Keras is on the MacBook Air with an M1 chip that has 8GB of unified memory.

**3.2.1 Federated learning.** FL is an architecture that revolutionizes the traditional DL approach unlike relying on data collection at a centralized location, it works in a distributed environment. The FL approach allows the participating clients to collaborate distributively. This architecture contains the participation of multiple clients and a single server, as shown in Fig 3. In contrast to DL, the FL doesn't store the data at the central location, rather, it keeps the data within clients. The datasets belonging to the clients are called local datasets. The server initiates the FL architecture by sending the global model trained using pre-defined weights to each client, which are further implemented at each participating client with their local datasets. The final parameters of each client are reiterated back to the server for aggregation. The FedAVG is utilized to collect the model parameters from the participating clients, and aggregation is performed. In the next phase, the aggregated parameters are sent to the server for updating the global model. This operation continues until the best and the most optimal model is generated. The proposed FL algorithm has been shown in Algorithm 1.

The proposed framework has N clients, each possessing its local glioma datasets denoted as $X_i$. These datasets have the labels denoted as $Y_i$. The i index has been used to represent the

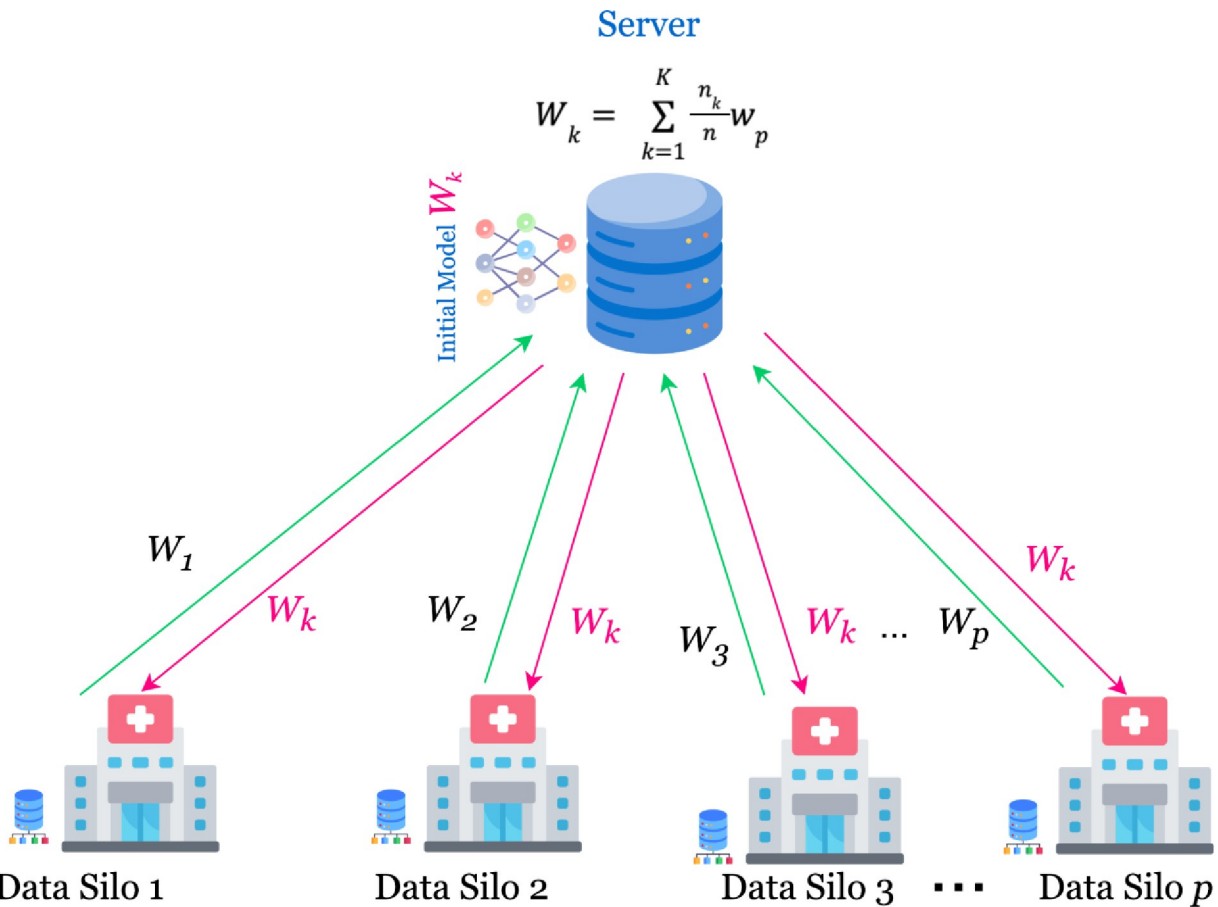

**Fig 3. A schematic architecture of federated learning.**

index of each client. The datasets contained within each participating client are denoted as $\{X_1, X_2, \ldots X_N\}$ having the respective labels represented as $\{Y_1, Y_2, \ldots Y_N\}$. The M is used to denote the global model, which is initially sent to each client to train itself with the local datasets. Further, the local parameters $\{M_1, M_2, \ldots M_N\}$ are parameterized with $w_i = W$. The $w_i$ and W are used for representing local and global parameters, respectively.

The updated parameters of each client are then aggregated using a FedAVG aggregation method as provided in Eq (1).

$$\underset{w}{minimize}\, F(w) = \frac{1}{F} \sum_{i=1}^{N} F_i(w) \tag{1}$$

Where, Fi(w) denotes the risk function for client each client {X1, X2, . . . XN}.

```
Algorithm 1: w: local updates; W: global model; r: communication
rounds; n: participating clients; Xt: random set of clients; each cli-
ent: n; W: global parameters; E: epochs at local clients; {W₁, W₂, ...
Wₙ}: local parameters; b: batch size; η: learning rate; Y: labels; X:
images; local model: datasets at clients: {X₁, X₂, …, Xₙ}; labels:
{Y₁, Y₂, ... Yₙ}; optimizer: Opt; local model: {M₁, M₂, …, Mₙ}; C: frac-
tion of dataset.
```

```
Server:
  initialize the model with w₀
  for round r = 1, 2, 3, ... do
    n ← max(C.K, 1)
    Xₜ ← choosing the random n clients
    for each client k ∈ Xt in parallel do
      wᵏₜ₊₂ ← ClientUpdates(k, wₜ)
    wₜ₊₂ ← ∑ᴷₖ₌₁ (pₖ/p) wᵏₜ₊₁
ClientUpdates(k, w):
  β ← (split the dataset Qₖ into batches with size B)
  for every local epoch a from 1 to E do
    for batch b ∈ β do
      w ← w η □ l(w;b)
    Return w to server
```

### 3.2.2 Proposed federated learning-based glioma brain tumour detection framework.

This section provides the details on the proposed methodology used for the development of privacy-protected FL-based glioma brain tumour detection model development. Fig 4 shows the proposed FL-based MobileNetV2 model for glioma brain tumour detection using 5-clients and 10-clients federated architecture. This model is highly privacy-preserved as the private data of the hospitals are neither shared with the peer clients nor with the central server. The initial global model in the central location has been distributed to each hospital, where the localized dataset is utilized to train the model at the hospital's location. This process remains decentralized as the data is never transferred to the central location or the parallel location of the participating clients. The proposed model is developed using a glioma dataset distributed among multiple clients, however, in practical scenarios, the participating hospitals have their own dataset for training the global model distributed by the central server. The dataset utilized for the training has been resized to 224X224 pixels, and data balancing has also been performed. The base DL model for FL-framework has been chosen as the MobileNetV2 model, which is a pre-trained neural network used for edge and mobile devices. It is the upgraded version of the MobileNet model that offers improved efficiency and performance. There are various key characteristics of the MobileNetV2 model, namely, inverted residuals, depthwise separable convolutions, linear bottlenecks, short connections, and efficient architecture. The depthwise separable convolutions are used to separate channel-wise and spatial convolutions, which reduces the number of computations and parameters, resulting in a better and improved model. The inverted residual with linear bottlenecks are the architectures that use lightweight linear bottlenecks among the layers. Linear bottlenecks contain three different operations: 1X1, convolution, depthwise convolution, and projection, which are used to maintain the balance between performance and model size. In addition, the shortcut connections are similar to residual blocks, which help correct the flow of model gradients between training phases. Lastly, efficient architectures are utilised for deploying the model on the devices to reduce the need for computational resources and time.

In the proposed work, this global MobileNetV2 model was initially trained using ImageNet weights. The model has been distributed among 5 and 10 clients in IID and non-IID data. The global model has been trained with the local datasets contained by 5 and 10 different clients, and the updated parameters at each client have been reiterated back to the server for aggregation. Before the updation, the weights from each client are collected and aggregated using the FedAVG technique and passed to the server for a global model update. This process continues until the optimal performance is achieved. The MobileNetV2 at the client's location is trained

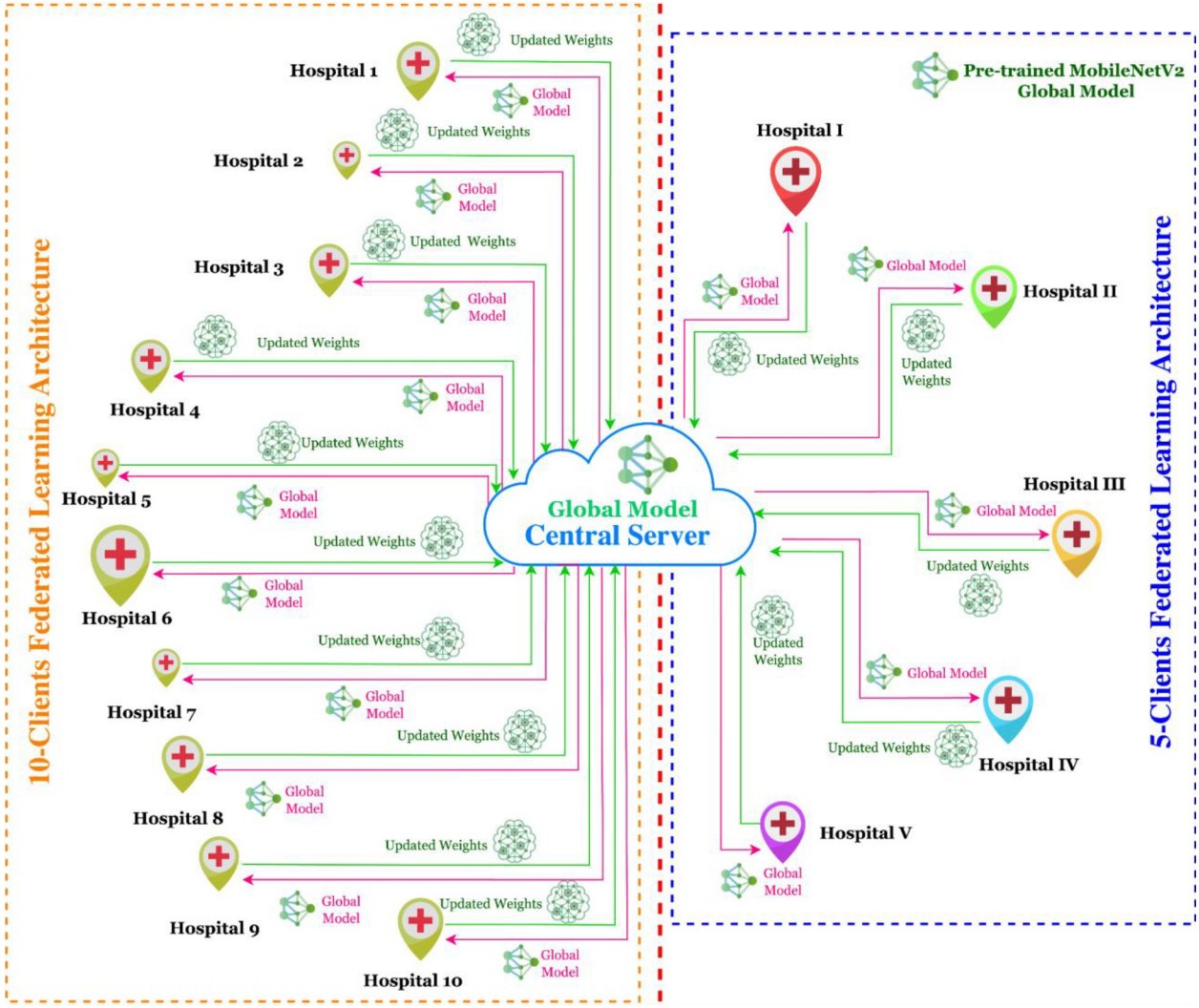

**Fig 4. Proposed FL-based MobileNetV2 model for glioma brain tumour detection using 5-clients and 10-clients federated architecture.**

for 10 epochs, and the global model is also updated 10 times in the entire architecture. The flow diagram of the proposed FL-based glioma detection model is shown in Fig 5. The Adam optimizer, batch size 32, learning rate at 0.001, and rounds of 10 have been configured to the model. Further, to identify the loss in binary classification, the Binary CrossEntropy function has been utilized. The model's hyperparameter configuration is provided in Table 3.

The FL model has been used to overcome the challenges of DL models. The FL frameworks are privacy-protected, scalable, more generalized, and cost-efficient, whereas the DL models have data security concerns. The DL models are centralized and require a large amount of datasets for achieving optimal results. The proposed work has been introduced for glioma detection in healthcare datasets, and medical information is sensitive and not easily accessible due to the strict privacy rules proposed by the HIPAA Act. Therefore, the FL architectures can be used over DL models to maintain data privacy and generalize improved outcomes in the healthcare sectors.

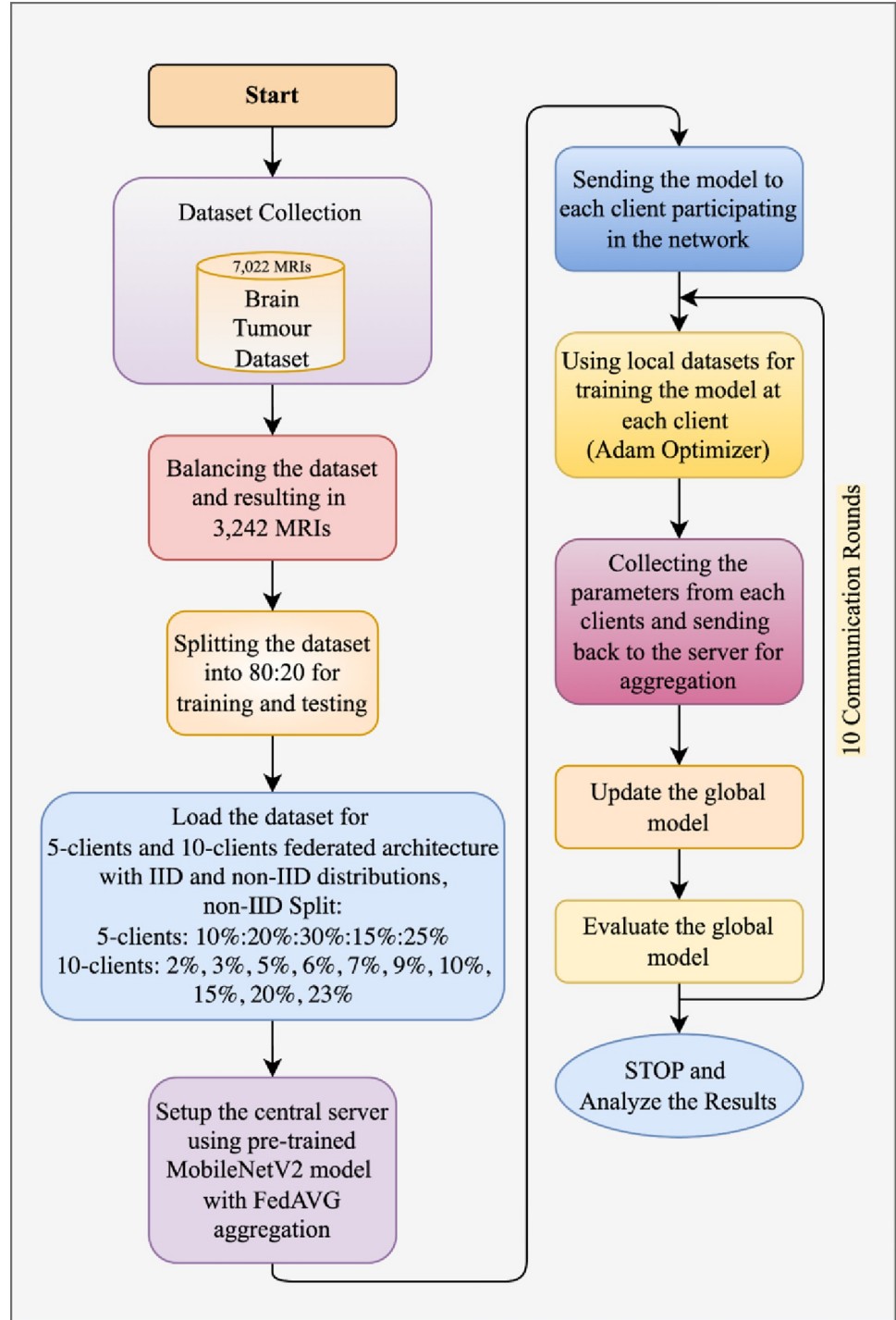

**Fig 5. Flow diagram of the proposed FL-based glioma detection model.**

## 4. Results and discussion

The proposed FL-based MobileNetV2 model has been implemented with Adam optimizer in 5 and 10 clients' federated architecture. The model's results have been analysed with both IID and non-IID data. This section discusses the results of the proposed model in 5 and 10 clients'

**Table 3. Hyperparameters configuration for the development of proposed federated learning-based glioma brain tumour detection framework.**

| Hyperparameters | Configuration | Hyperparameters | Configuration |
|---|---|---|---|
| Learning Rate | 0.001 | Rounds | 10 |
| Batch Size | 32 | Clients | 5 and 10 |
| Optimizer | Adam | Update Aggregation | FedAVG |
| Local Epochs | 10 | Client Utilization | 100% |
| Global Epochs | 10 | Loss Metric | Binary CrossEntropy |

FL-framework with IID and non-IID data distributions in terms of overall and client-wise accuracy, recall, precision, and F1-score as shown in Eqs (2), (3), (4), and (5).

$$\text{Accuracy} = \frac{C}{A} * 100 \tag{2}$$

Where,

C is the samples correctly recognized.

A is the total number of samples.

$$\text{Recall} = \frac{true\ positives}{true\ positives + false\ negatives} \tag{3}$$

Where,

*true positives* are the actual positive instances correctly predicted by the model.

*false negatives* are the actual negative instances wrongly predicted by the model.

$$\text{Precision} = \frac{true\ positives}{true\ positives + false\ positives} \tag{4}$$

Where,

*true positives* are the actual positive instances correctly predicted by the model.

*false positives* are the actual positive instances wrongly predicted by the model.

$$\text{F1-score} = \frac{2*Precision*Recall}{Precision + Recall} \tag{5}$$

## 4.1 The results of the proposed FL-based MobileNetV2 model implemented with Adam optimizer for 5 clients in IID distribution

This section incorporates the performance outcomes of the proposed FL model in 5 clients with IID data.

Fig 6 illustrates the accuracy of the proposed model in 5 clients' architecture with IID data. The graph shows that the highest accuracy of 99.76% and a minimum accuracy of 90.87% have resulted in rounds 9 and round 1, respectively. The accuracy values are highly consistent and show the similarity between actual and predicted outcomes.

Fig 7 shows the precision, recall, and F1-score of the proposed model in 5 clients architecture with IID data. The values of precision at different rounds are nearly close to 1, which indicates that the model is highly capable of achieving optimal outcomes due to minimum variability among the achieved values. The recall of the proposed model in 5 clients FL-framework with IID data. The recall at the initial round resulted in 0.9049, which increased after round 1 and reached 0.9970 at round 9. This graph depicts that the model has been improved over rounds and is capable of classifying accurate outcomes. The F1-score of the proposed model in 5 clients' FL-framework with IID data. The F1-score at the initial round resulted in

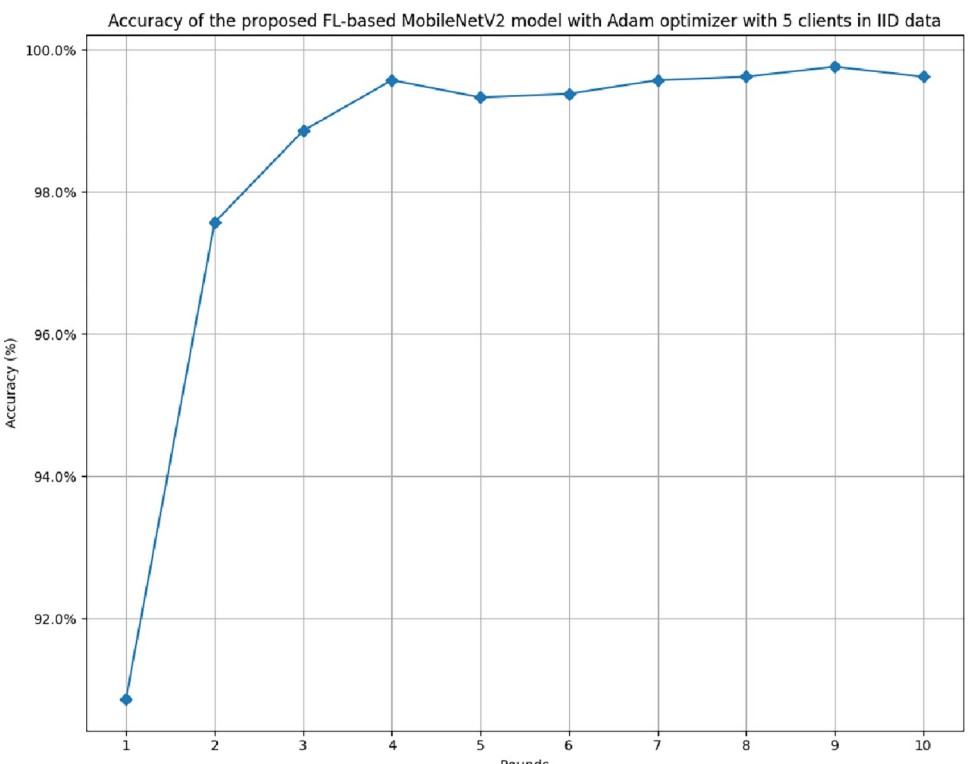

**Fig 6. Accuracy of the proposed model in 5 clients' FL-framework with IID data.**

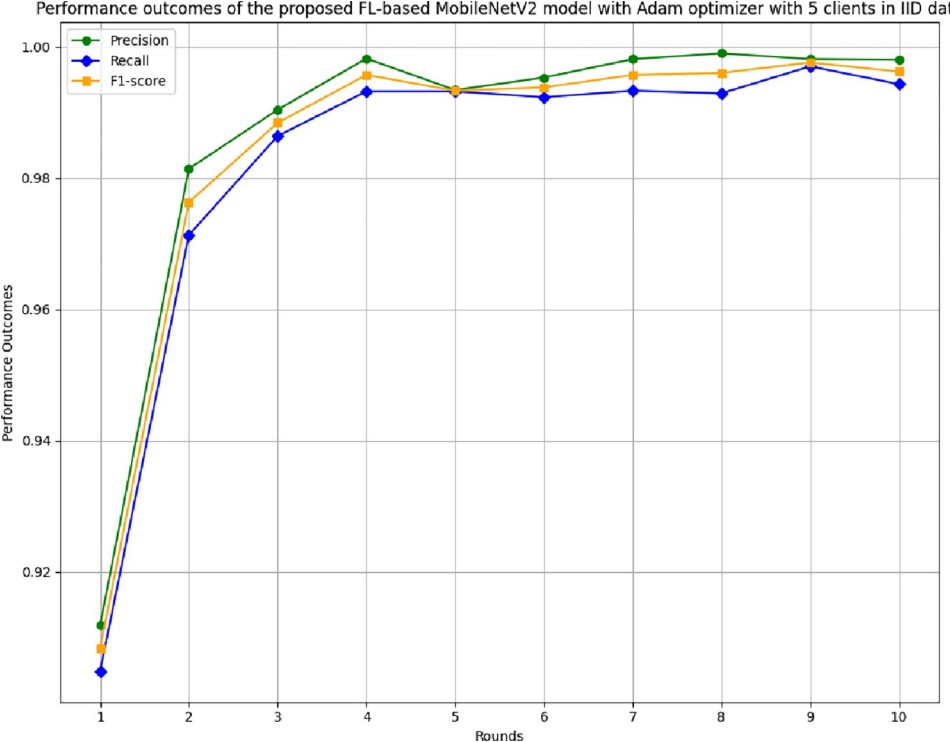

**Fig 7. Precision, recall, and F1-score of the proposed model in 5 clients' FL-framework with IID data.**

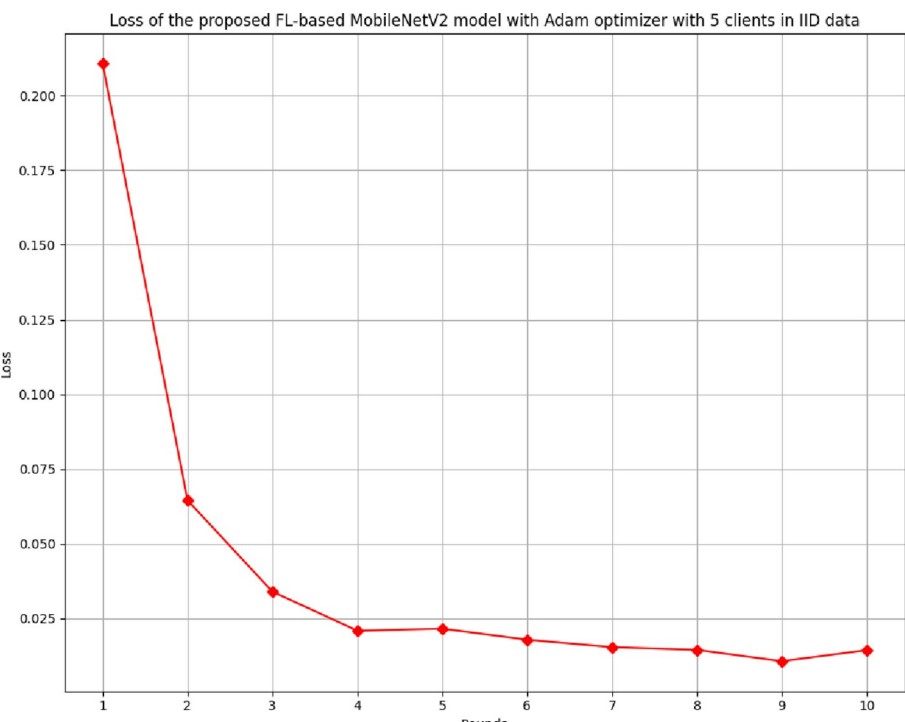

**Fig 8. Loss of the proposed model in 5 clients' FL-framework with IID data.**

0.9084 depicting a better performance from the initial round itself, however, with the round value increase, the F1-score started to progress and reached 0.9976 at round 9, resulting in improved model generalization and prediction.

Fig 8 depicts the loss of the FL-centric model in 5 clients' FL-framework with IID data. The lowest loss of 0.0107 has been identified at round value 9. The value of the loss at each round shows a gradual decrease with the enhancement in round value.

Table 4 provides the round-wise performance outcomes of 5 clients' federated framework with IID data.

Fig 9 depicts the round-wise accuracy of 5 different clients in a FL model with IID data. The accuracy of client 1 has been started with 92.79%, which reached 99.76%, whereas clients 2, 3, 4, and 5 have achieved 100%, 99.76%, 99.76%, and 100% accuracy, respectively. The performance of each client has been identified as improved over rounds, which results in a better generalization of the predicted outcomes.

Fig 10 shows the round-wise recall of 5 clients in FL-framework with IID data. The minimum recall of clients 1 to 5 has been started with 0.9375, 0.8857, 0.9162, 0.8529, and 0.9322, which reached 1.00, 1.00, 0.9947, 0.995, and 1.00, respectively. The recall graph of each client shows a continuous improvement in the values with the rounds resulting in better learning and optimal performance.

Fig 11 depicts the round-wise precision of 5 different clients in the FL-framework with IID data. The highest precision at each client 1, 2, 3, 4, and 5 have resulted as 1.00, whereas this value started from the initial value of 0.9292, 0.8732, 0.9021, 0.911, and 0.9442, respectively.

**Table 4. A detailed predictive analysis of round-wise performance outcomes of 5 clients federated framework with IID data.**

| Metrics | C1 | C2 | C3 | C4 | C5 |
|---|---|---|---|---|---|
| Accuracy | Minimum: 92.79% at round 1<br>Maximum: 99.76% at rounds 6, 8, and 10 | Minimum: 87.74% at round 1<br>Maximum: 100% at rounds 8 and 10 | Minimum: 91.59% at round 1<br>Maximum: 99.76% at rounds 8 and 10 | Minimum: 88.70% at round 1<br>Maximum: 99.76% at rounds 7 and 10 | Minimum: 93.56% at round 1<br>Maximum: 100% at rounds 7 and 8 |
| Recall | Minimum: 0.9375 at round 1<br>Maximum: 1.0000 at round 8 | Minimum: 0.8857 at round 1<br>Maximum: 1.0000 at rounds 4, 8, and 10 | Minimum: 0.9162 at round 1<br>Maximum: 0.9947 at rounds 8, 9, and 10 | Minimum: 0.8529 at round 1<br>Maximum: 0.9950 at rounds 7 and 10 | Minimum: 0.9322 at round 1<br>Maximum: 1.0000 at rounds 4, 7, and 8 |
| Precision | Minimum: 0.9292 at round 1<br>Maximum: 1.0000 at rounds 6, 7, 9, and 10 | Minimum: 0.8732 at round 1<br>Maximum: 1.0000 at rounds 8 and 10 | Minimum: 0.9021 at round 1<br>Maximum: 1.0000 at rounds 2, 4, 6, 7, and 8 | Minimum: 0.9110 at round 1<br>Maximum: 1.0000 at rounds 4, 7, 9, and 10 | Minimum: 0.9442 at round 1<br>Maximum: 1.0000 at rounds 4, 7, and 8 |
| F1-score | Minimum: 0.9333 at round 1<br>Maximum: 0.9977 at rounds 7 and 8 | Minimum: 0.8794 at round 1<br>Maximum: 1.0000 at rounds 8 and 10 | Minimum: 0.9091 at round 1<br>Maximum: 0.9973 at round 8 | Minimum: 0.8810 at round 1<br>Maximum: 0.9975 at rounds 7 and 10 | Minimum: 0.9382 at round 1<br>Maximum: 1.0000 at rounds 4, 7, and 8 |
| Loss | Minimum: 0.0044 at round 7<br>Maximum: 0.1764 at round 1 | Minimum:0.0030 at round 8<br>Maximum: 0.2429 at round 1 | Minimum:0.0130 at rounds 9 and 10<br>Maximum: 0.2067 at round 1 | Minimum:0.0034 at round 6<br>Maximum: 0.2560 at round 1 | Minimum:0.0012 at round 7<br>Maximum: 0.1709 at round 1 |

Fig 12 shows the F1-score of 5 different clients in FL-framework with IID data. The highest F1-score achieved by clients 1 to 5 are 0.9977, 1.00, 0.9973, 0.9975, and 1.00, respectively, where these F1-scores were identified as 0.9333, 0.8794, 0.9091, 0.881, and 0.9382 at round 1, respectively.

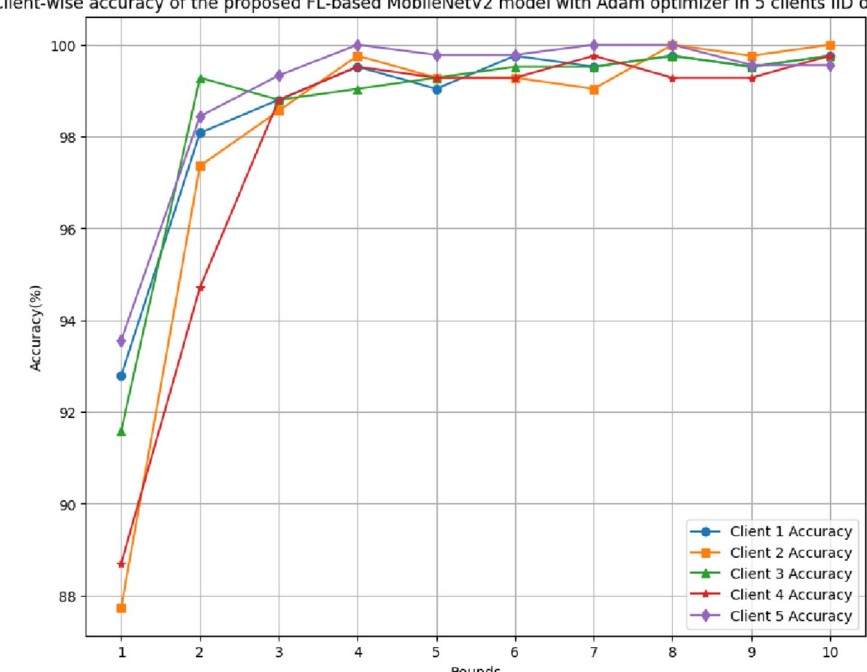

**Fig 9. Round-wise accuracy of 5 clients in FL-framework with IID data.**

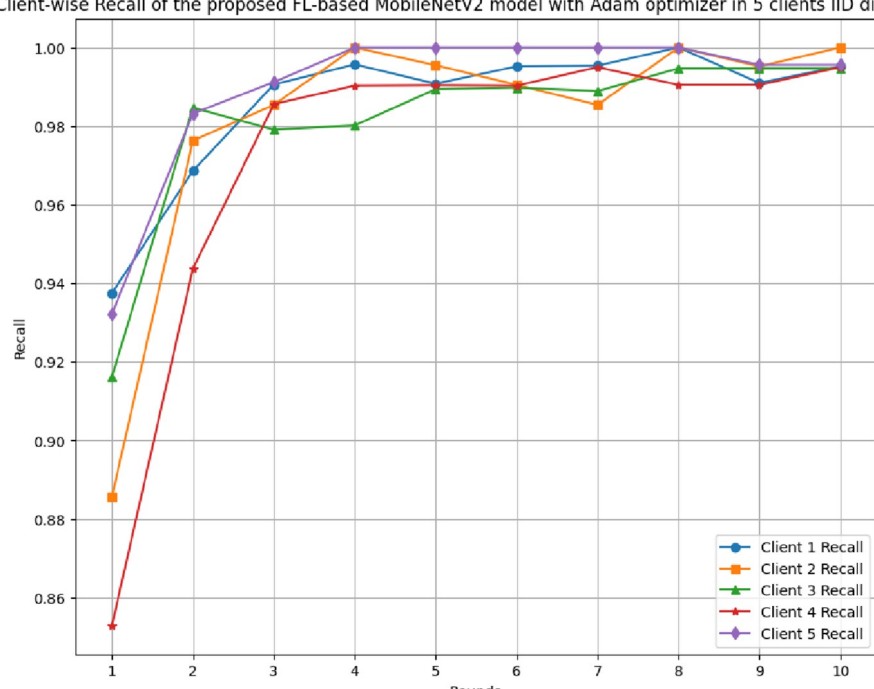

**Fig 10. Round-wise recall of 5 clients in FL-framework with IID data.**

Fig 13 illustrates the round-wise loss of 5 different clients in FL-framework with IID data. The highest and lowest loss of 0.1764 and 0.0044, 0.2429 and 0.0030, 0.2067 and 0.0130, 0.2560 and 0.0034, and 0.1709 and 0.0012, respectively.

## 4.2 The results of the proposed FL-based MobileNetV2 model implemented with Adam optimizer for 10 clients in IID distribution

This section discusses the performance outcomes of the proposed FL model in 10 client architecture with IID data.

Fig 14 shows the accuracy of the proposed model in 10 clients FL-framework with IID data. The model has shown the highest accuracy at 99.64% at round value 10, whereas the accuracy started with the value of 77.77% at round 1. This accuracy has been identified as improving over rounds, which shows that the model becomes highly optimal when it has been trained with increasing epoch values.

Fig 15 depicts the recall, precision, and F1-score of the proposed model in 10 clients' FL-framework with IID data. The highest recall was achieved in round 10 at 0.9954, whereas the lowest recall at 0.8056 was initiated in round 1. The precision of the proposed model in 10 clients FL-framework with IID data. The highest precision of 0.9973 has been resulted by round 8. The precision value has been identified as continuously improving over rounds increase, resulting in better generalization of the results and optimal model development. The F1-score of the proposed model in 10 clients' FL-framework with IID data. The highest F1-score of 0.9959 has resulted at round 10 depicting a balance between recall and precision values.

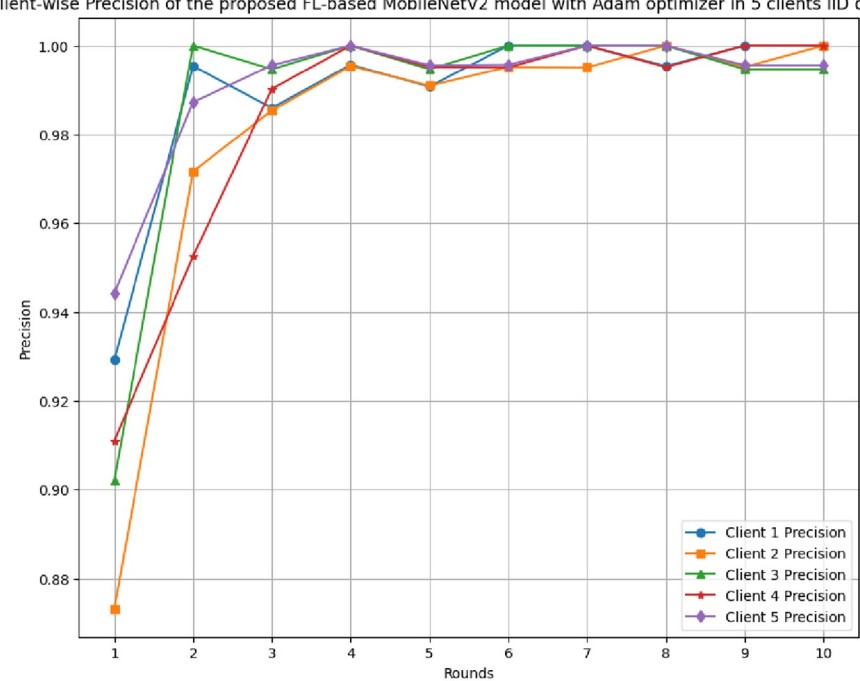

**Fig 11. Round-wise precision of 5 clients in FL-framework with IID data.**

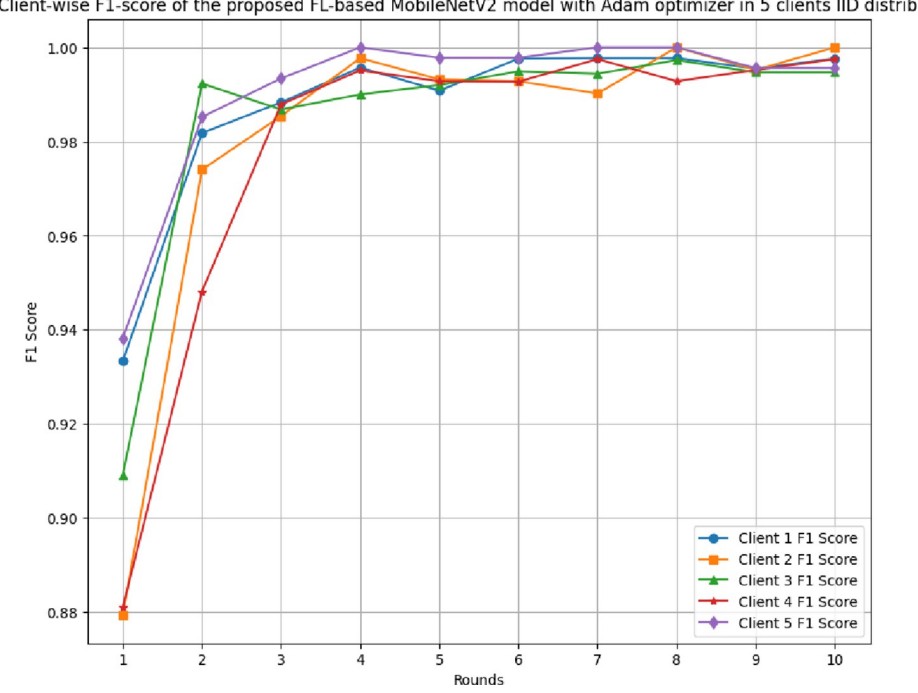

**Fig 12. Round-wise F1-score of 5 different clients in FL-framework with IID data.**

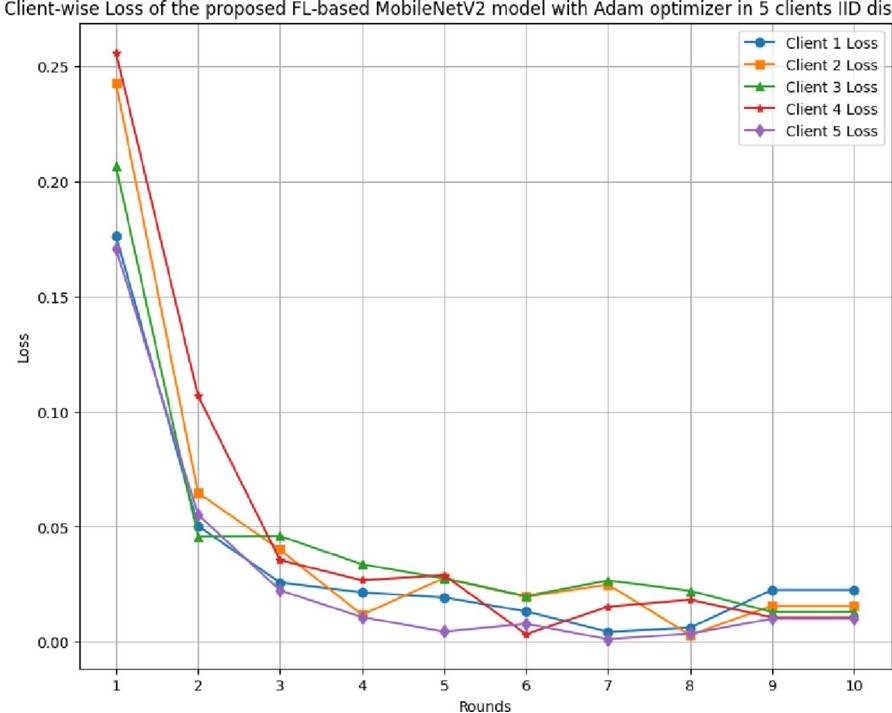

**Fig 13. Round-wise loss of 5 different clients in FL-framework with IID data.**

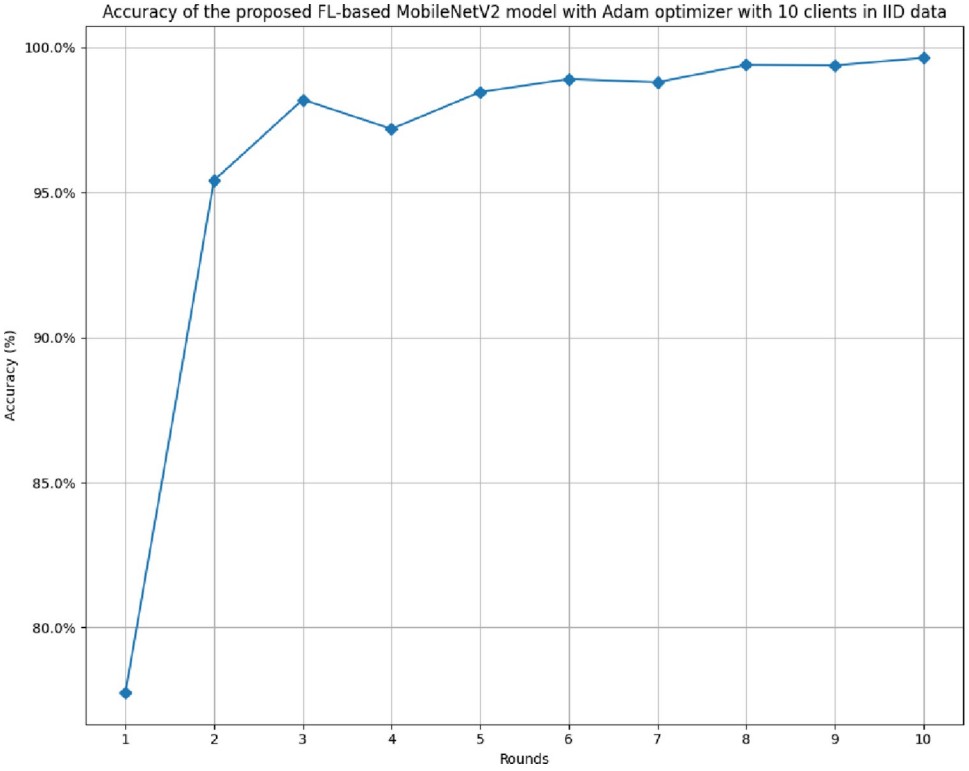

**Fig 14. Accuracy of the proposed model in 10 clients' FL-framework with IID data.**

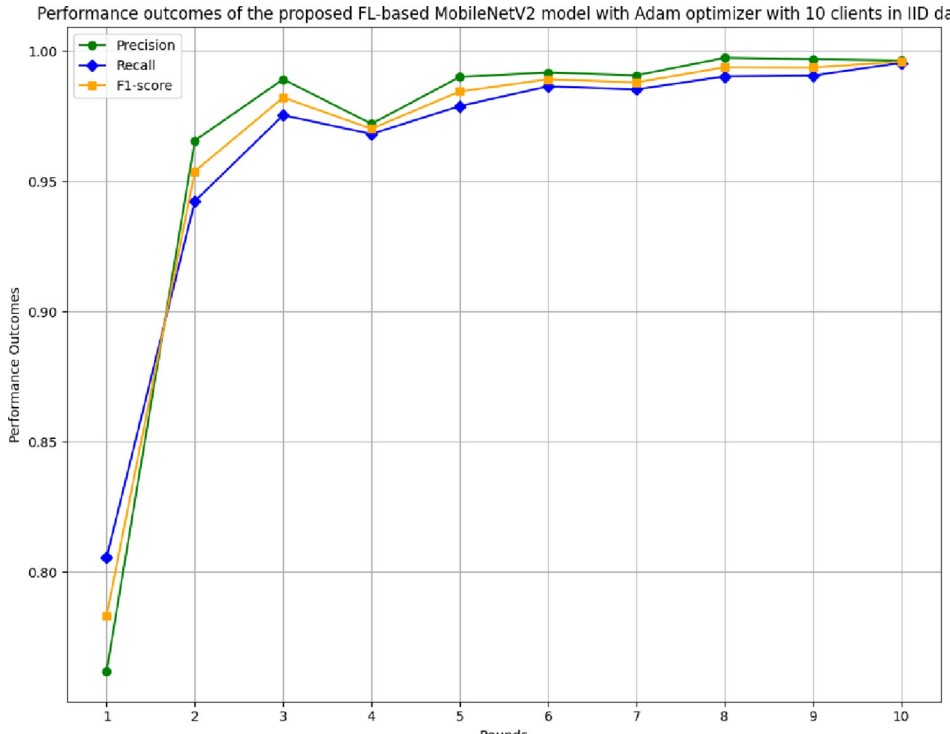

**Fig 15. Precision, recall, and F1-score of the proposed model in 10 clients FL-framework with IID data.**

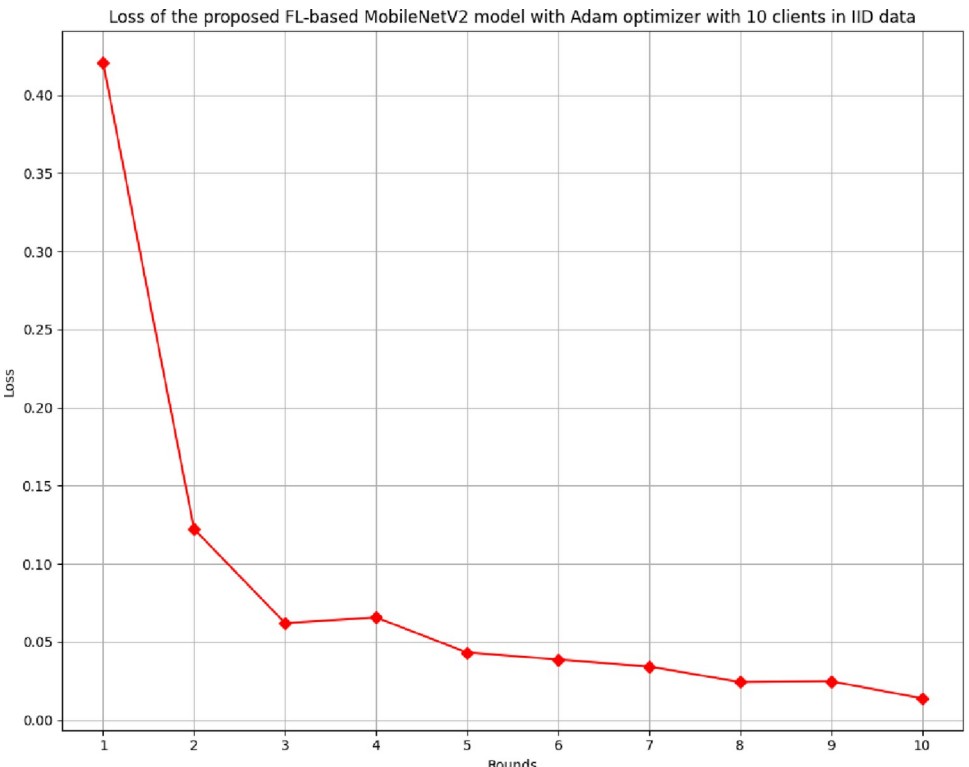

**Fig 16. Loss of the proposed model in 10 clients FL-framework with IID data.**

**Table 5. A detailed predictive analysis of round-wise performance outcomes of 10 clients federated framework with IID data.**

| Metrics | C1 | C2 | C3 | C4 | C5 | C6 | C7 | C8 | C9 | C10 |
|---|---|---|---|---|---|---|---|---|---|---|
| **Accuracy** | Minimum: 79.69% at round 1 Maximum: 100% at rounds 6 and 100 | Minimum: 79.69 at round 1 Maximum: 100% at round 10 | Minimum: 76.04% at round 1 Maximum: 100% at round 10 | Minimum: 81.77% at round 1 Maximum: 100% at round 10 | Minimum: 71.88% at round 1 Maximum: 100% at rounds 6 and 10 | Minimum: 70.83% at round 1 Maximum: 100% at round 6 | Minimum: 76.56% at round 1 Maximum: 100% at round 10 | Minimum: 82.29% at round 1 Maximum: 100% at rounds 76, 7, and 10 | Minimum: 68.75% at round 1 Maximum: 100% at rounds 6 and 10 | Minimum: 90.16% at round 1 Maximum: 100% at rounds 6 and 10 |
| **Recall** | Minimum: 0.8776 at round 1 Maximum: 1.0000 at rounds 6 and 10 | Minimum: 0.7905 at round 1 Maximum: 1.0000 at rounds 6, 7, 8, 9, and 10 | Minimum: 0.8108 at round 1 Maximum: 1.0000 at rounds 5 and 10 | Minimum: 0.8627 at round 1 Maximum: 1.0000 at round 10 | Minimum: 0.8000 at round 1 Maximum: 0.9881 at round 10 | Minimum: 0.7011 at round 1 Maximum: 1.0000 at round 4 | Minimum: 0.7882 at round 1 Maximum: 1.0000 at round 10 | Minimum: 0.7397 at round 1 Maximum: 1.0000 at rounds 6, 7, and 10 | Minimum: 0.7670 at round 1 Maximum: 1.0000 at round 10 | Minimum: 0.9183 at round 1 Maximum: 1.0000 at rounds 6 and 10 |
| **Precision** | Minimum: 0.7611 at round 1 Maximum: 1.0000 at rounds 8, 9, and 10 | Minimum: 0.8300 at round 1 Maximum: 1.0000 at rounds 5 and 10 | Minimum: 0.7826 at round 1 Maximum: 1.0000 at rounds 3, 6, 7, 8, 9, and 10 | Minimum: 0.8073 at round 1 Maximum: 1.0000 at rounds 3, 4, and 10 | Minimum: 0.6847 at round 1 Maximum: 0.9881 at round 10 | Minimum: 0.6703 at round 1 Maximum: 1.0000 at round 10 | Minimum: 0.7128 at round 1 Maximum: 1.0000 at rounds 6, 8, and 9 | Minimum: 0.7826 at round 1 Maximum: 1.0000 at rounds 2, 6, 8, 9, and 10 | Minimum: 0.6870 at round 1 Maximum: 1.0000 at rounds 5, 6, 7, 8, 9, and 10 | Minimum: 0.9009 at round 1 Maximum: 1.0000 at rounds 6, 7, and 10 |
| **F1-score** | Minimum: 0.8152 at round 1 Maximum: 1.000 at round 10 | Minimum: 0.8098 at round 1 Maximum: 1.000 at round 10 | Minimum: 0.7964 at round 1 Maximum: 1.000 at round 10 | Minimum: 0.8341 at round 1 Maximum: 1.000 at round 10 | Minimum: 0.7379 at round 1 Maximum: 0.9881 at round 10 | Minimum: 0.6854 at round 1 Maximum: 0.9891 at round 10 | Minimum: 0.7486 at round 1 Maximum: 0.9874 at round 10 | Minimum: 0.7605 at round 1 Maximum: 1.000 at round 10 | Minimum: 0.7248 at round 1 Maximum: 1.000 at round 10 | Minimum: 0.9095 at round 1 Maximum: 1.000 at round 10 |
| **Loss** | Maximum: 0.3453 at round 1 Minimum: 0.0073 at round 10 | Maximum: 0.3736 at round 1 Minimum: 0.0071 at round 10 | Maximum: 0.4384 at round 1 Minimum: 0.0069 at round 10 | Maximum: 0.3750 at round 1 Minimum: 0.0070 at round 10 | Maximum: 0.5384 at round 1 Minimum: 0.0363 at round 10 | Maximum: 0.5753 at round 1 Minimum: 0.0145 at round 10 | Maximum: 0.3538 at round 1 Minimum: 0.0378 at round 10 | Maximum: 0.3582 at round 1 Minimum: 0.0112 at round 10 | Maximum: 0.6442 at round 1 Minimum: 0.0070 at round 10 | Maximum: 0.2032 at round 1 Minimum: 0.0038 at round 10 |

Fig 16 shows the loss of the proposed model in 10 clients FL-framework with IID data. The lowest loss of 0.0139 was identified at round 10, whereas the highest loss was analyzed at round 1.

Table 5 tabulates the round-wise performance outcomes of 10 clients federated framework with IID data.

Fig 17 shows the round-wise accuracy of 10 clients in FL-framework with IID data. The highest accuracy of each client ranging from client 1 to client 10 has been analyzed as 100%, whereas the least accuracy was identified as 79.69%, 79.69%, 76.04%, 81.77%, 71.88%, 80.83%, 76.56%, 82.29%, 68.75%, and 90.16%, respectively. The improvement in accuracy over rounds shows that the model is highly efficient optimal performance due to the least variance between actual and predicted outcomes.

Fig 18 shows the round-wise recall of 10 different clients in FL-framework with IID data. The highest recall of clients 1, 2, 3, 4, 6, 7, 8, 9, and 10 have resulted as 1.00, whereas client 5 has a 0.9881 recall value. The improvement in recall value shows that the model becomes more generalized with the increase in round values.

Fig 19 depicts the precision of 10 different clients in FL-framework with IID data. The clients involved in this architecture have performed optimally, whereas client 5 has resulted in the highest precision value of 0.9881.

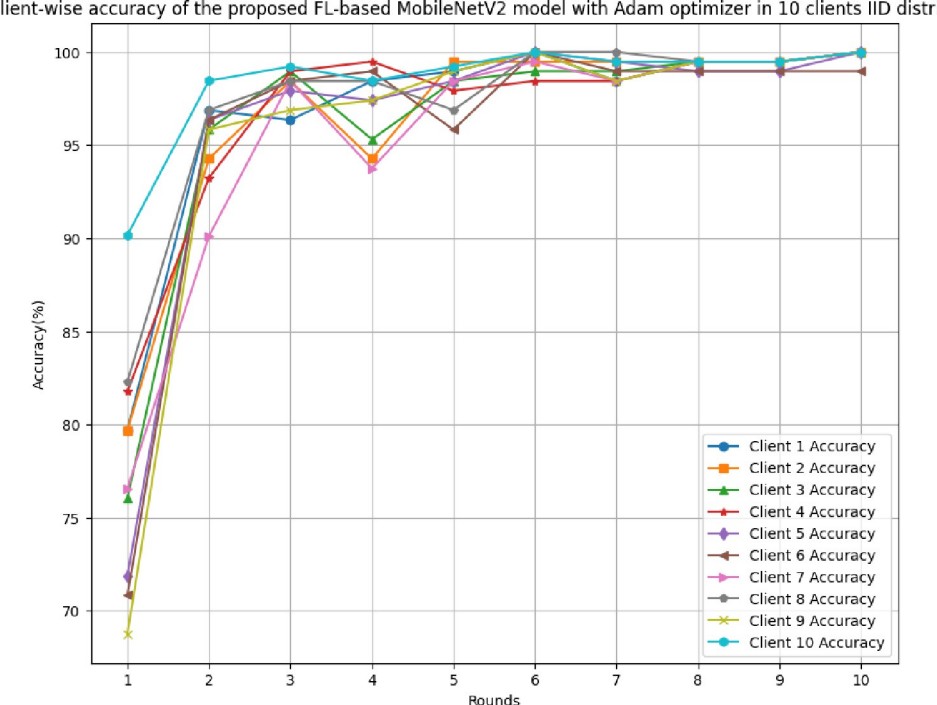

**Fig 17. Round-wise accuracy of 10 different clients in FL-framework with IID data.**

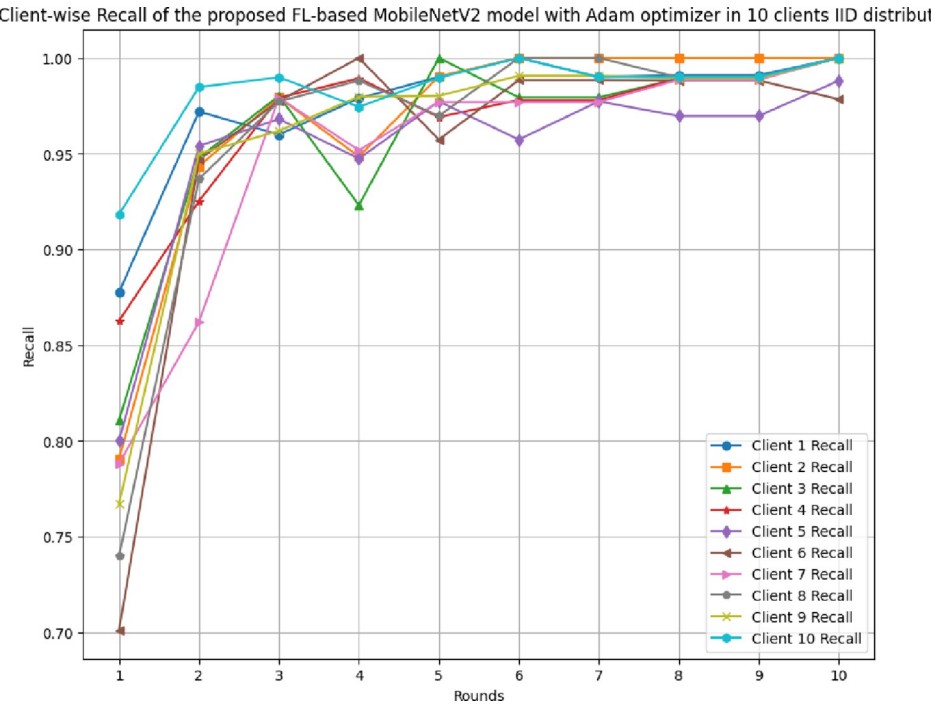

**Fig 18. Round-wise recall of 10 clients in FL-framework with IID data.**

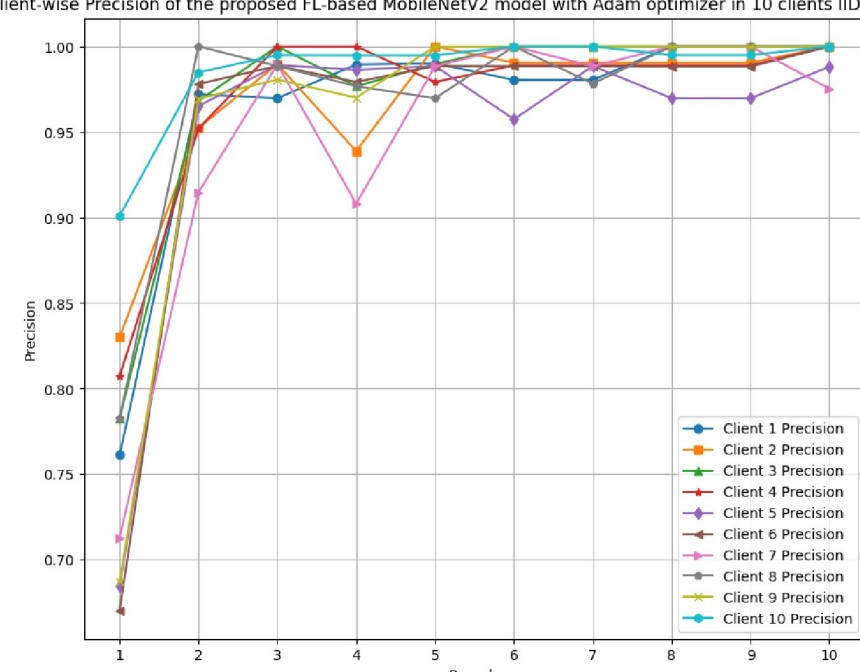

**Fig 19. Round-wise precision of 10 different clients in FL-framework with IID data.**

Fig 20 shows the round-wise F1-score of 10 different clients in FL-framework with IID data. The highest F1-score for clients 1, 2, 3, 4, 8, 9, and 10 have been identified as 1.00, whereas the clients 5, 6, and 7 have resulted as 0.9881, 0.9891, and 0.9874, respectively.

Fig 21 illustrates the round-wise loss of 10 different clients in the FL model with IID data. The highest loss has been resulted by client 9 as 0.6442, whereas the lowest loss of 0.0038 has been identified with client 10. The initial value of loss also resulted in the least in client 10 at 0.2032. Though, the model has an improved performance, however, client 10 was resulted as the most optimal in generalizing the prediction outcomes.

### 4.3 The results of the proposed FL-based MobileNetV2 model implemented with Adam optimizer for 5 clients in non-IID distribution

This section discusses the performance outcomes of the proposed model in 5 clients' FL-framework with non-IID data.

Fig 22 shows the accuracy of the proposed model in 5 clients FL-framework with non-IID data. The accuracy trend line has been identified as improving over rounds and the highest accuracy of 99.71% has been achieved as round value 6. This improvement shows that the model is highly efficient in terms of predicting the outcomes with the least error value.

Fig 23 depicts the precision, recall, and F1-score of the proposed model in 5 clients FL-framework with non-IID data. This model achieved the highest precision of 1.00 at round 6, which indicates that the predicted outcomes are highly precise, resulting in optimal model development. The recall of the proposed model in 5 clients' FL-framework with non-IID data. Round 5 has been identified as the most optimal recall-achieving round, with a value of 0.9962. The F1-score of the proposed model in 5 clients FL with non-IID data. The highest F1-score value of 0.998 resulted in round 6, whereas the minimum F1-score resulted in round

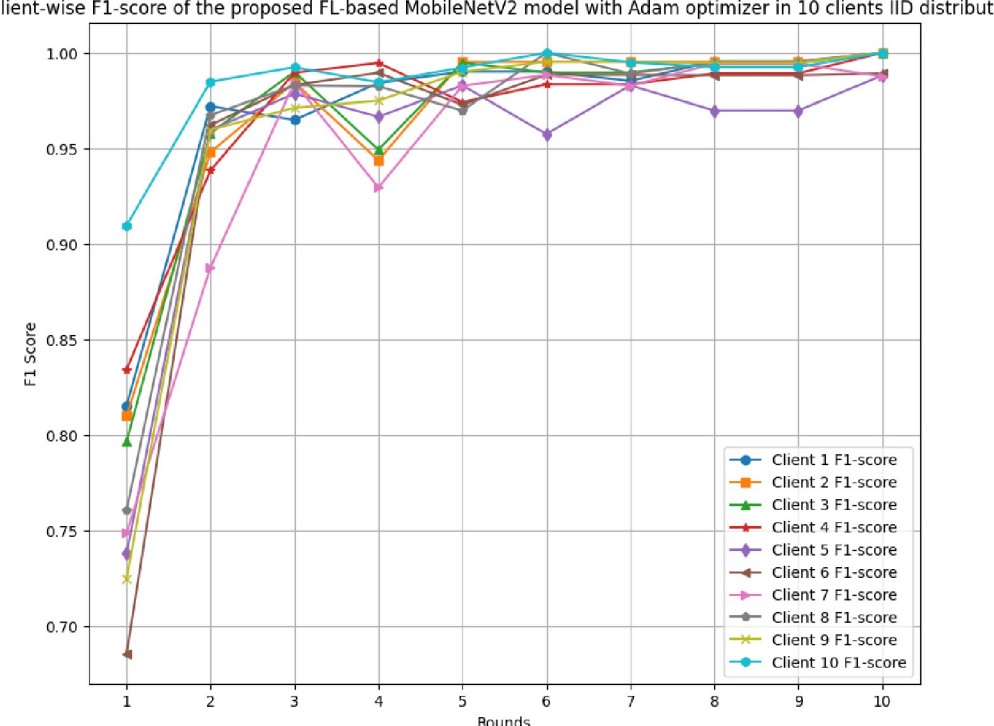

**Fig 20. Round-wise F1-score of 10 different clients in FL-framework with IID data.**

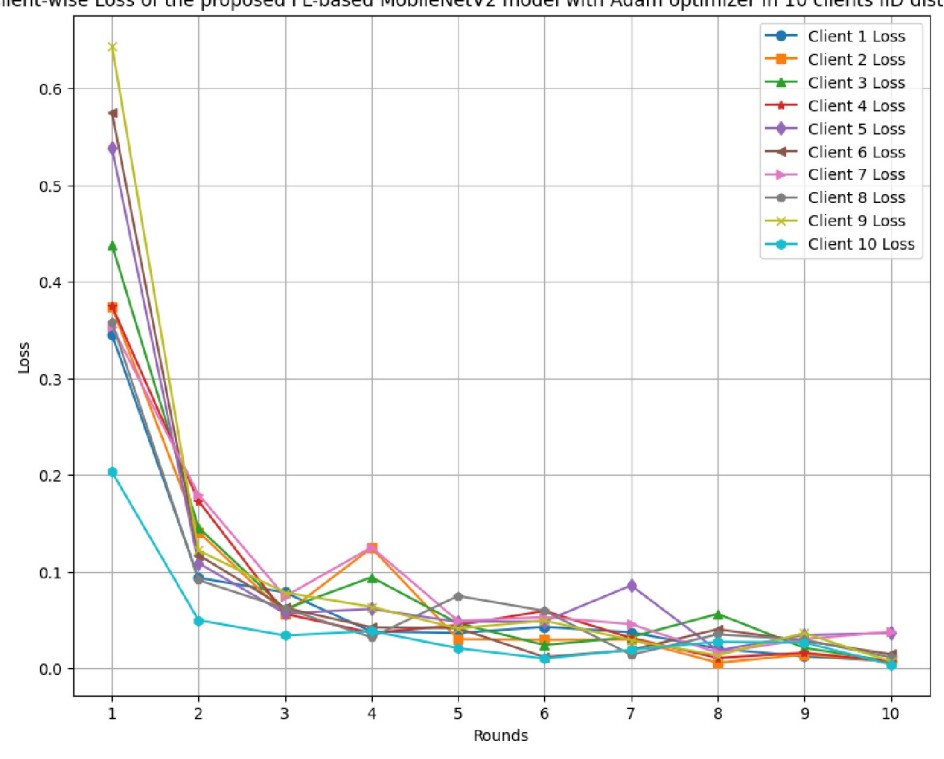

**Fig 21. Round-wise loss of 10 different clients in FL-framework with IID data.**

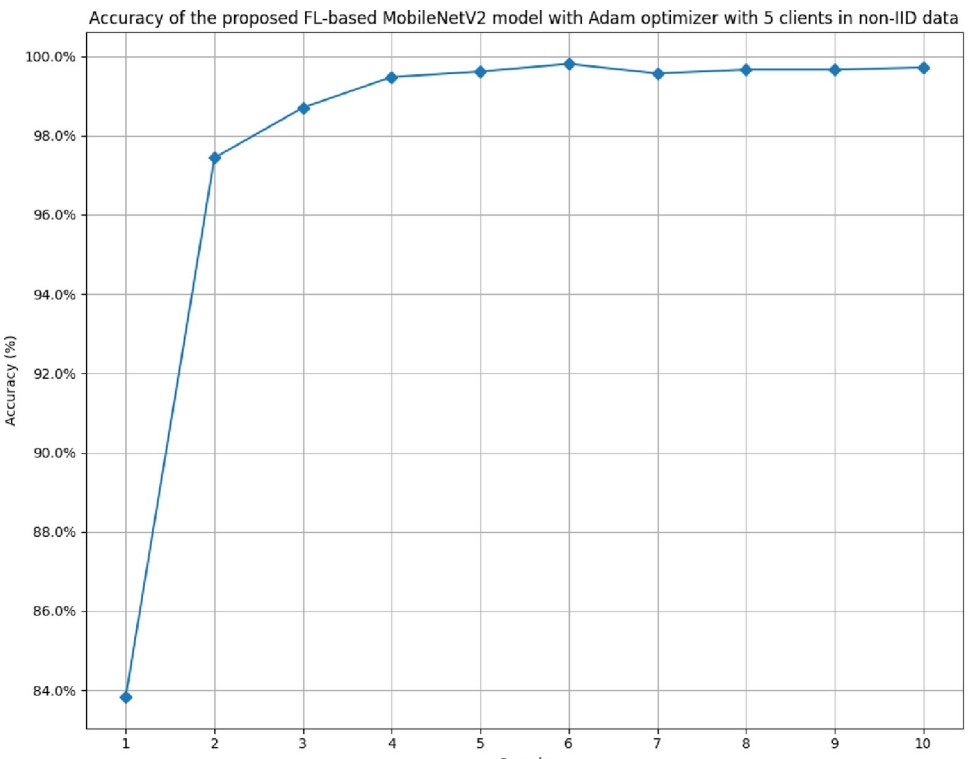

**Fig 22. Accuracy of the proposed model in 5 clients' FL-framework with non-IID data.**

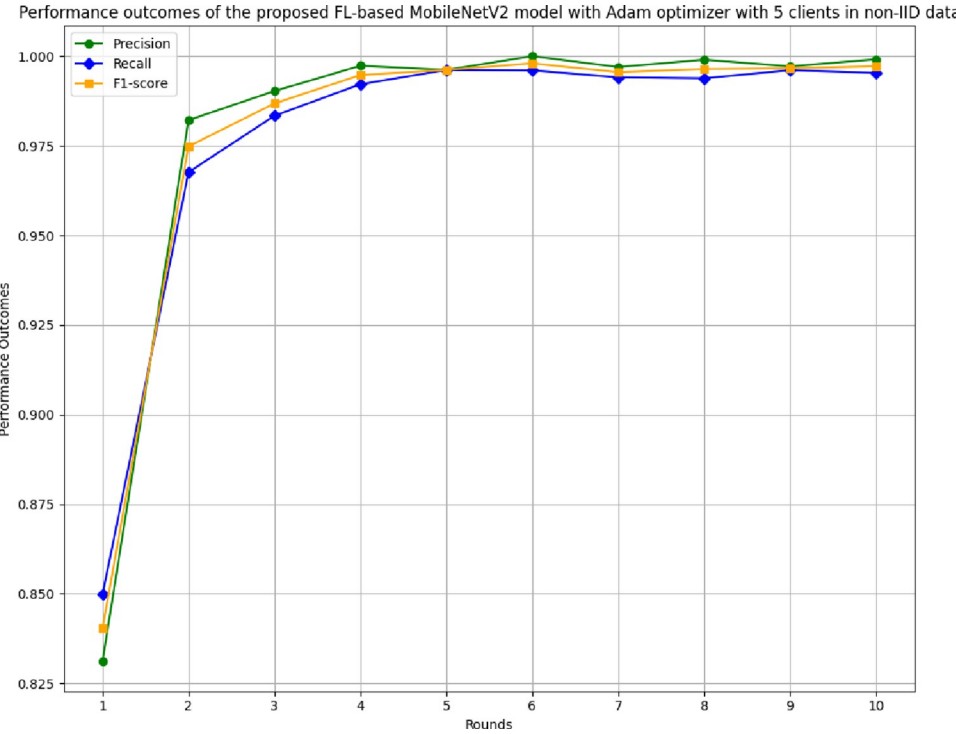

**Fig 23. Precision, recall, and F1-score of the proposed model in 5 clients FL-framework with non-IID data.**

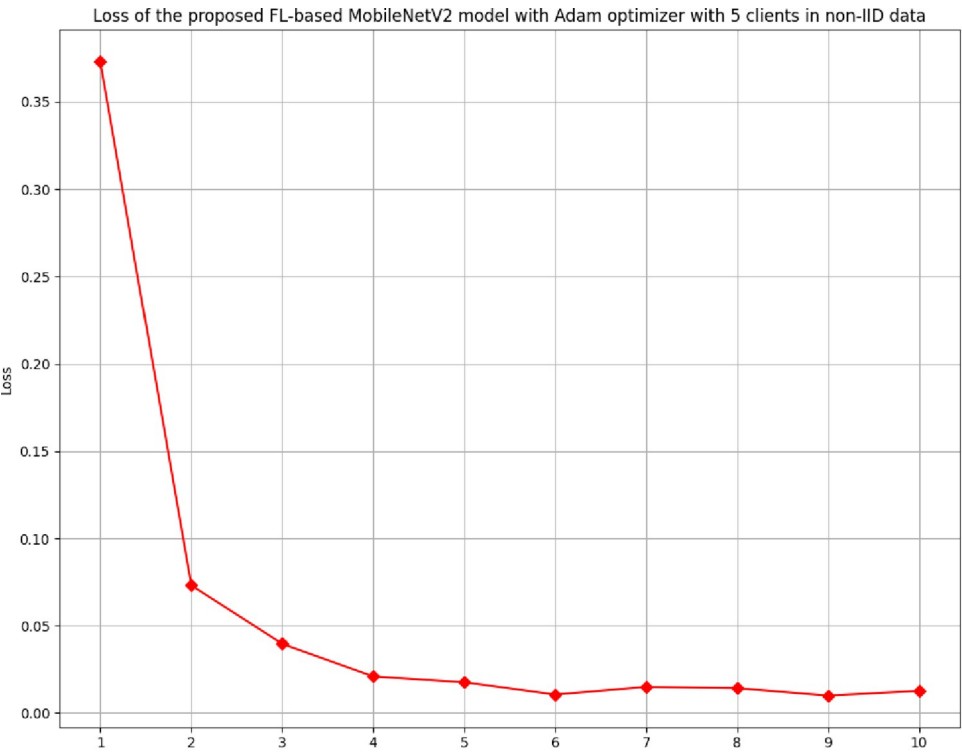

**Fig 24. Loss of the proposed model in 5 clients' FL-framework with non-IID data.**

1. The F1-score trend line shows that the model performance has been significantly improved over rounds, however, after round 6, the performance started to degrade.

Fig 24 depicts the loss of the proposed model in 5 clients FL-framework with non-IID data. The lowest loss has been identified as round 6, with a value of 0.0107.

Table 6 tabulates the round-wise performance outcomes of 5 clients federated framework with non-IID data.

Fig 25 shows the round-wise accuracy of 5 different clients in the FL model with non-IID data. The highest accuracy among all the clients has been resulted by clients 1, 3, 4, and 5 as 100%, whereas the least value of highest accuracy has been identified by client 2 as 99.76%.

The round-wise recall of 5 different clients in FL-framework with non-IID data has been shown in Fig 26. The highest recall among all participating clients has been achieved as 1.00 with all the clients. However, the initial lowest recall has been identified with client 3 as 0.7958 at round 1.

Fig 27 shows the round-wise precision of 5 different clients in FL-framework with non-IID data. The highest precision value of 1.00 was identified by each client, whereas the lowest starting precision of 0.79 was achieved by client 2. The precision value at each client shows that the model is highly optimal for better results generalization.

Fig 28 depicts the round-wise F1-score of 5 different clients in FL-framework with non-IID data. The highest F1-score of 1.00 has been resulted by clients 1, 3, 4, and 5, whereas the lowest F1-score of 0.9977 has been resulted by client 2.

Fig 29 illustrates the round-wise F1-score of 5 different clients in FL-framework with non-IID data. The loss of clients 1, 2, 3, 4, and 5 have been resulted as 0.0040, 0.0053, 0.0028,

**Table 6. A detailed predictive analysis of round-wise performance outcomes of 5 clients federated framework with non-IID data.**

| Metrics | C1 | C2 | C3 | C4 | C5 |
|---------|----|----|----|----|----|
| **Accuracy** | Minimum: 83.65% at round 1 Maximum: 100.0% at rounds 6 and 8 | Minimum: 80.05% at round 1 Maximum: 99.76% at rounds 9 and 10 | Minimum: 81.01% at round 1 Maximum: 100.0% at round 10 | Minimum: 88.46% at round 1 Maximum: 100.0% at round 6 | Minimum: 86.0% at round 1 Maximum: 100.0% at rounds 5, 6, 7, 8, and 9 |
| **Recall** | Minimum: 0.8661 at round 1 Maximum: 1.0000 at rounds 4, 5, 6, 8, and 10 | Minimum: 0.8238 at round 1 Maximum: 1.0000 at round 9 | Minimum: 0.7958 at round 1 Maximum: 1.0000 at round 10 | Minimum: 0.8824 at round 1 Maximum: 1.0000 at round 6 | Minimum: 0.8814 at round 1 Maximum: 1.0000 at rounds 4, 5, 6, 7, 8, 9, and 10 |
| **Precision** | Minimum: 0.8362 at round 1 Maximum: 1.0000 at rounds 5, 6, 8, 9, and 10 | Minimum: 0.7900 at round 1 Maximum: 1.0000 at rounds 6 and 10 | Minimum: 0.7917 at round 1 Maximum: 1.0000 at rounds 4, 6, 8, 9, and 10 | Minimum: 0.8824 at round 1 Maximum: 1.0000 at rounds 4, 5, 6, 7, 8, and 10 | Minimum: 0.8560 at round 1 Maximum: 1.0000 at rounds 3, 5, 6, 7, 8, and 9 |
| **F1-score** | Minimum: 0.8509 at round 1 Maximum: 1.0000 at rounds 5, 6, 8, and 10 | Minimum: 0.8065 at round 1 Maximum: 0.9977 at round 9 | Minimum: 0.7937 at round 1 Maximum: 1.0000 at round 10 | Minimum: 0.8824 at round 1 Maximum: 1.0000 at round 6 | Minimum: 0.8685 at round 1 Maximum: 1.0000 at rounds 5, 6, 7, 8, and 9 |
| **Loss** | Minimum: 0.0040 at round 8 Maximum: 0.3829 at round 1 | Minimum: 0.0053 at round 9 Maximum: 0.4529 at round 1 | Minimum: 0.0028 at round 10 Maximum: 0.4210 at round 1 | Minimum: 0.0062 at round 6 Maximum: 0.2750 at round 1 | Minimum: 0.0018 at round 8 Maximum: 0.3330 at round 1 |

0.0062, and 0.0018, respectively. These losses at the highest values have resulted in 0.3829, 0.4529, 0.421, 0.275, and 0.333, respectively, for clients 1, 2, 3, 4, and 5.

## 4.4 The results of the proposed FL-based MobileNetV2 model implemented with Adam optimizer for 10 clients in non-IID distribution

This section discusses the results of the proposed FL-based MobileNetV2 model implemented with Adam optimizer for 10 clients in non-IID distribution.

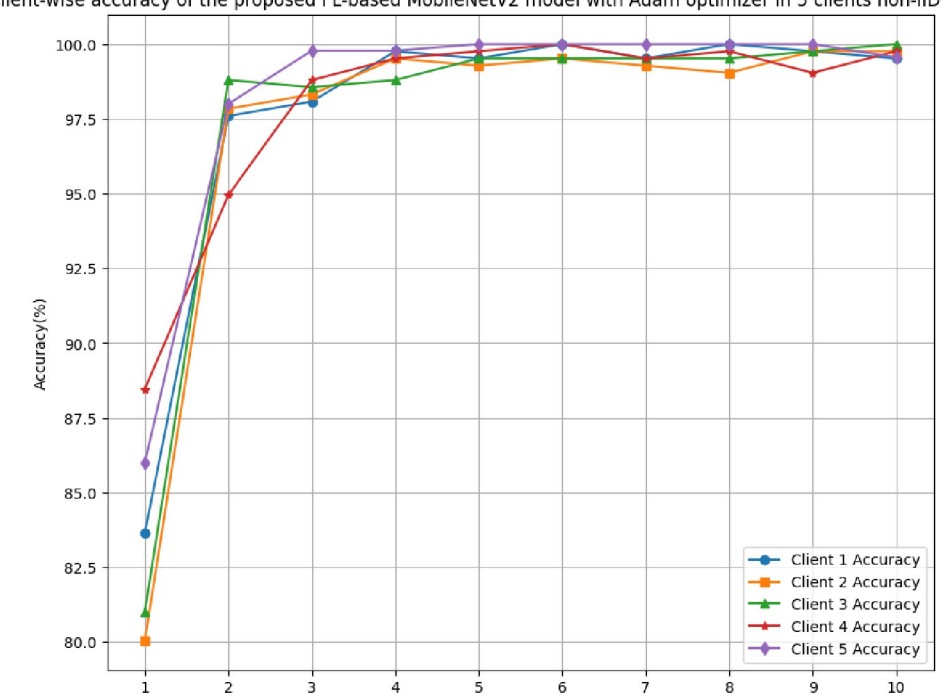

**Fig 25. Round-wise accuracy of 5 different clients in FL-framework with non-IID data.**

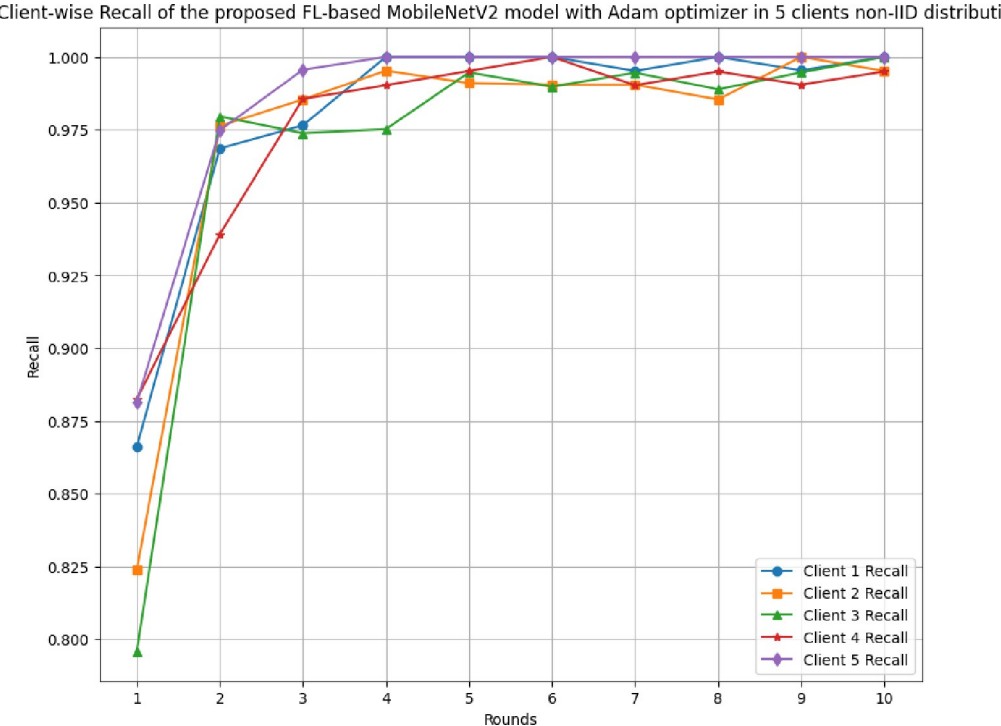

**Fig 26. Round-wise recall of 5 different clients in FL-framework with non-IID data.**

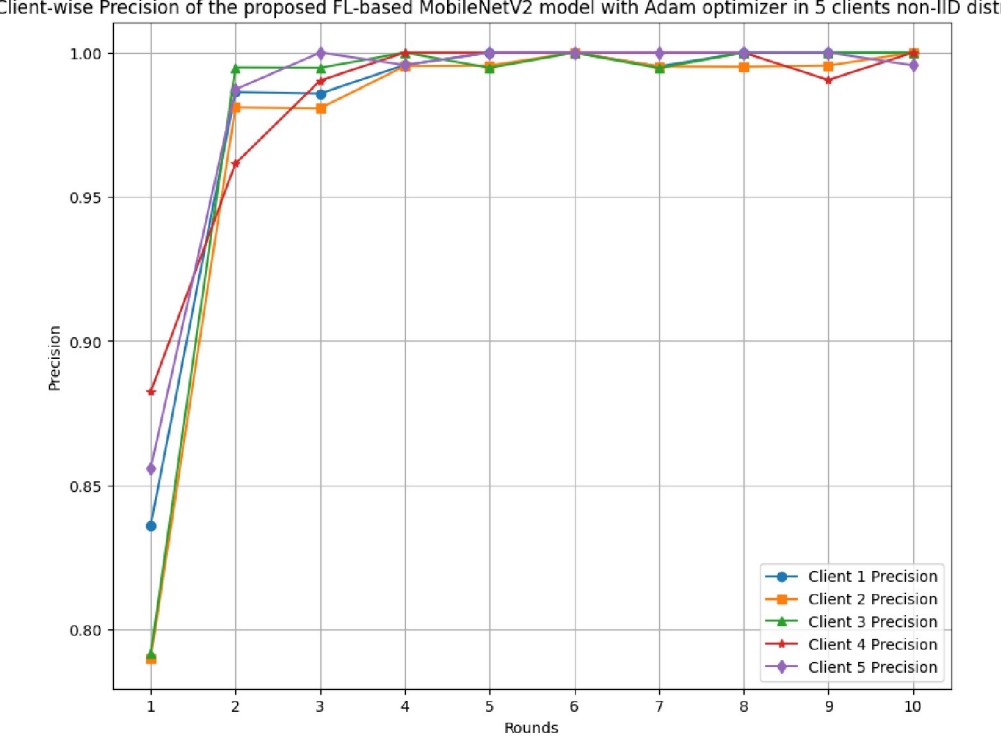

**Fig 27. Round-wise precision of 5 different clients in FL-framework with non-IID data.**

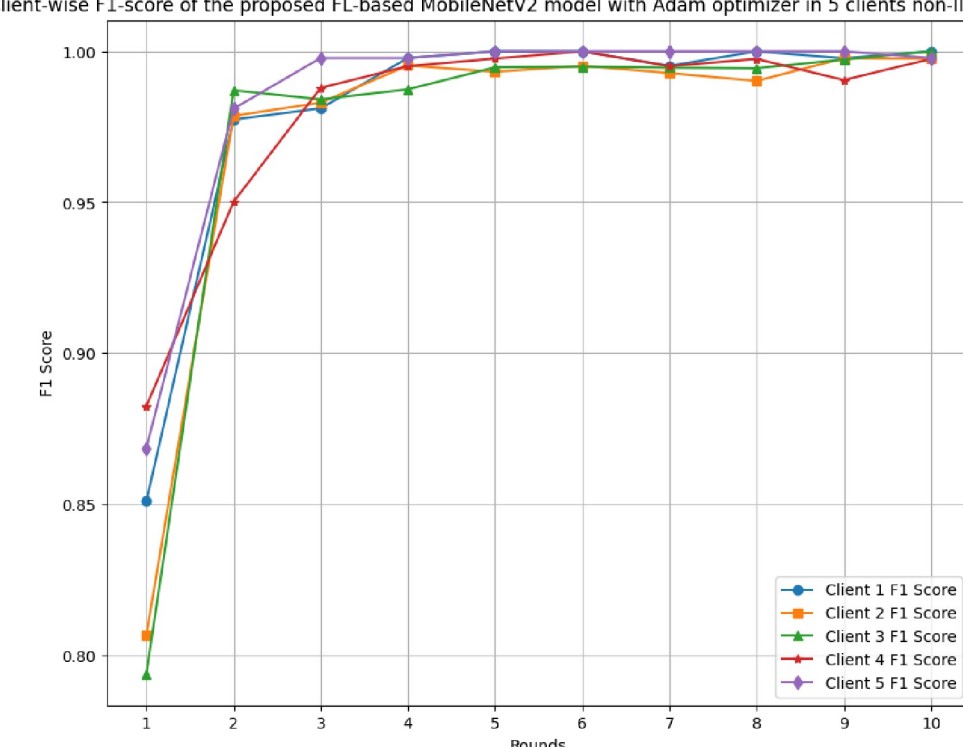

**Fig 28. Round-wise F1-score of 5 different clients in FL-framework with non-IID data.**

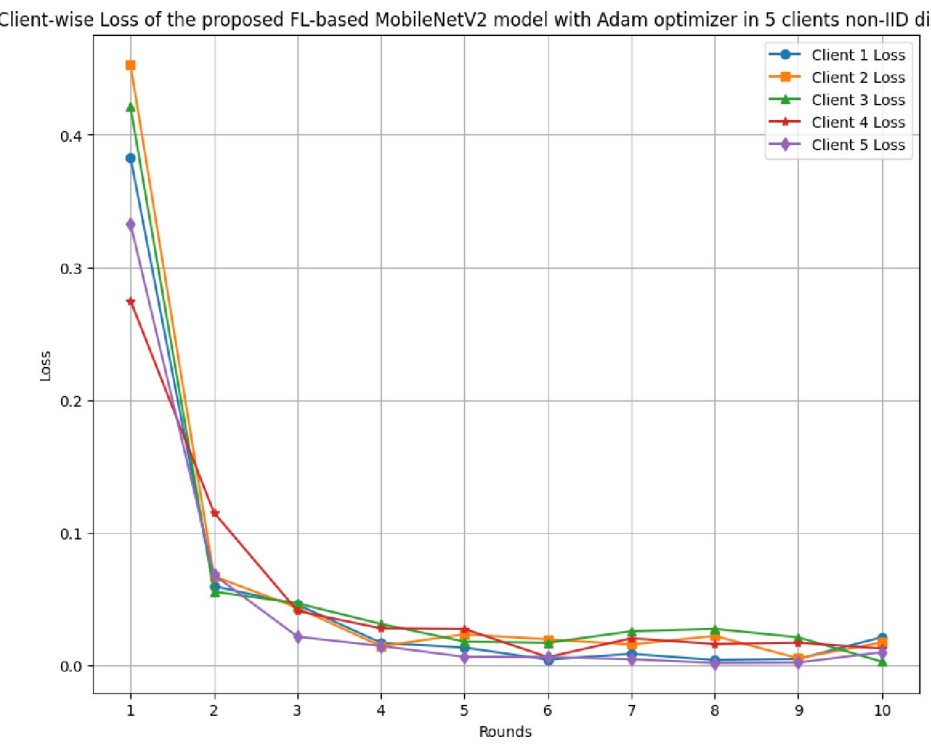

**Fig 29. Round-wise loss of 5 different clients in FL-framework with non-IID data.**

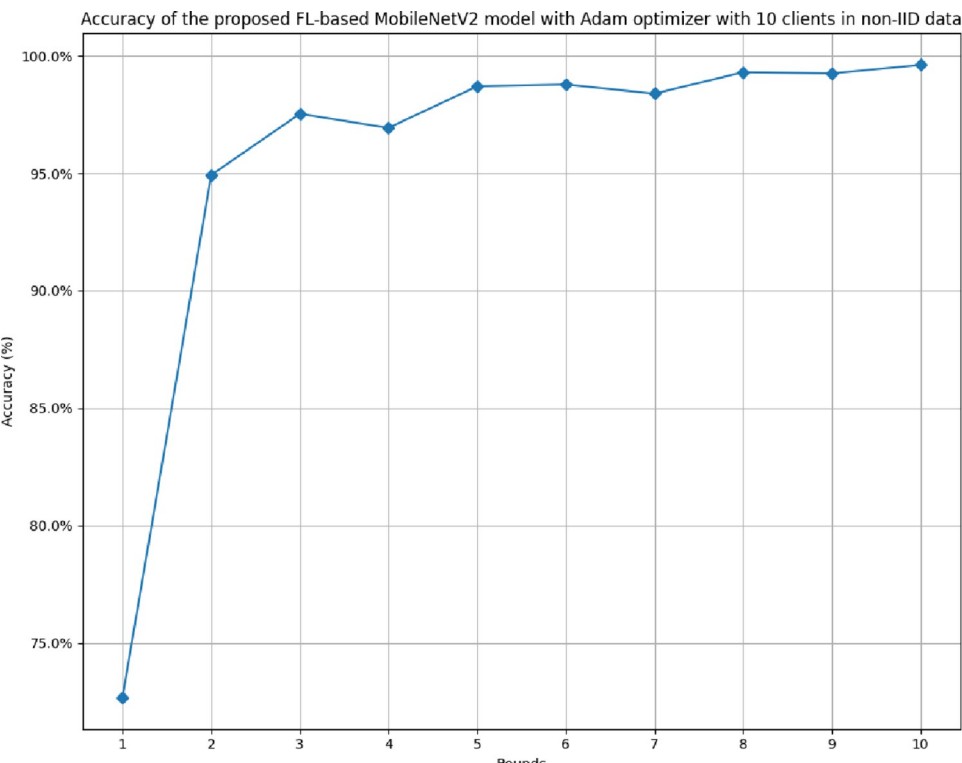

**Fig 30. Accuracy of the proposed model in 10 clients FL-framework with non-IID data.**

Fig 30 illustrates the accuracy of the proposed model in 10 clients architecture with non-IID data. The highest accuracy, 99.61%, was achieved at round 10, whereas the lowest accuracy was achieved at round 1. The accuracy of the Adam optimizer with non-IID data has been found to significantly improve over round values.

Fig 31 shows the precision, recall, and F1-score of the proposed model in 10 clients FL-framework with non-IID data. The highest precision of 0.9995 has been resulted by round 8. The improvement in the precision value has been analyzed, which shows that the model is highly precise for the accurate prediction of glioma brain tumour cases. The highest recall of 0.9928 has been resulted in round 8. The F1-score of the proposed model in 10 clients FL-framework with non-IID data. The highest F1-score of 0.9995 was identified in round 8, and the lowest F1-score was obtained in round 1.

The loss of the proposed model in 10 clients FL-framework with non-IID data has been shown in Fig 32. The lowest loss of the model has been resulted at round 10 as 0.0188. The value of loss depicts that the model is highly applicable of predicting accurate outcomes with the least error rate.

Table 7 tabulates the round-wise performance outcomes of 10 clients federated framework with non-IID data.

Fig 33 shows round-wise accuracy of 10 different clients in FL-framework with non-IID data. The clients involved in this framework has been significantly improved the accuracy over rounds. However, clients 1, 2, 3, 4, 6, 7, and 8 haves outperformed by resulting in an accuracy value of 100%, whereas, clients 5, 9, and 10 have resulted in 99.48%, 99.48%, and 99.74% accuracies, respectively.

Fig 34 depicts the round-wise recall of 10 different clients in FL-framework with non-IID data. The highest recall among all the clients have been resulted by clients 1, 2, 3, 4, 6, and 8.

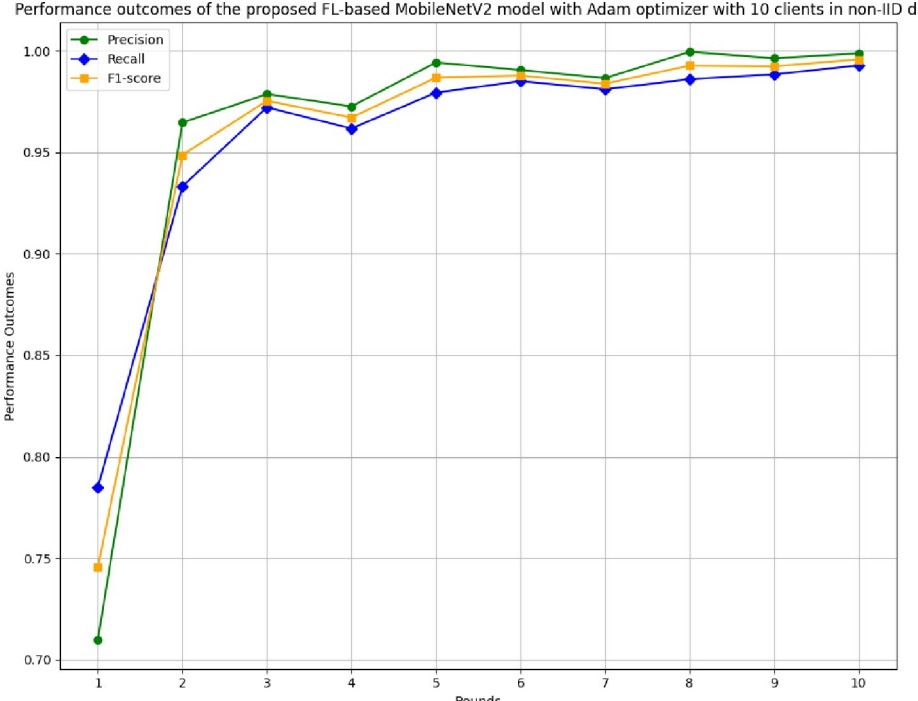

**Fig 31. Precision, recall, and F1-score of the proposed model in 10 clients FL-framework with non-IID data.**

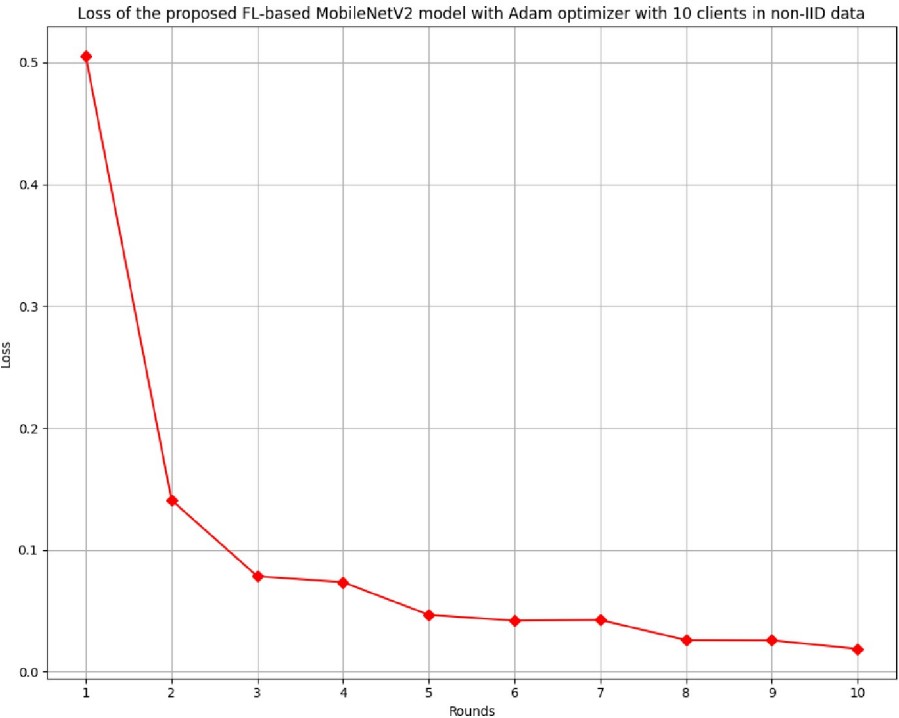

**Fig 32. Loss of the proposed model in 10 clients FL-framework with non-IID data.**

**Table 7. A detailed predictive analysis of round-wise performance outcomes of 10 clients federated framework with non-IID data.**

| Metrics | C1 | C2 | C3 | C4 | C5 | C6 | C7 | C8 | C9 | C10 |
|---|---|---|---|---|---|---|---|---|---|---|
| **Accuracy** | Minimum: 73.44% at round 1 Maximum: 100% at round 10 | Minimum: 69.79% at round 1 Maximum: 100% at rounds 8 and 10 | Minimum: 66.15% at round 1 Maximum: 100% at round 9 | Minimum: 72.92%at round 1 Maximum: 100% at round 10 | Minimum: 65.62%at round 1 Maximum: 99.48% at rounds 8 and 10 | Minimum: 79.17% at round 1 Maximum: 100% at round 6 | Minimum: 70.83% at round 1 Maximum: 100% at round 5 | Minimum: 83.85% at round 1 Maximum: 100% at rounds 7 and 10 | Minimum: 63.02% at round 1 Maximum: 99.48% at rounds 8, 9, and 10 | Minimum: 82.12%at round 1 Maximum: 99.74% at rounds 6 and 10 |
| **Recall** | Minimum: 0.7755 at round 1 Maximum: 1.00 at round 10 | Minimum: 0.7714 at round 1 Maximum: 1.00 at rounds 7, 8, 9, and 10 | Minimum: 0.8198 at round 1 Maximum: 1.00 at round 9 | Minimum: 0.8039 at round 1 Maximum: 1.00 at round 10 | Minimum: 0.7579 at round 1 Maximum: 0.9889 at round 8 | Minimum: 0.7931 at round 1 Maximum: 1.00 at round 6 | Minimum: 0.7529 at round 1 Maximum: 0.9901 at round 8 | Minimum: 0.7671 at round 1 Maximum: 1.00 at rounds 7 and 10 | Minimum: 0.7573 at round 1 Maximum: 0.9902 at round 8 | Minimum: 0.8510 at round 1 Maximum: 0.995 at rounds 5 and 6 |
| **Precision** | Minimum: 0.7238 at round 1 Maximum: 1.00 at rounds 5, 8, 9, and 10 | Minimum: 0.7043 at round 1 Maximum: 1.00 at rounds 8 and 10 | Minimum: 0.6691 at round 1 Maximum: 1.00 at rounds 8, 9, and 10 | Minimum: 0.7193 at round 1 Maximum: 1.00 at rounds 4, 7, 8, 9, and 10 | Minimum: 0.6261 at round 1 Maximum: 1.00 at rounds 5, 8, and 10 | Minimum: 0.7582 at round 1 Maximum: 1.00 at rounds 6, 8, 9, and 10 | Minimum: 0.6465 at round 1 Maximum: 1.00 at rounds 8, 9, and 10 | Minimum: 0.8000 at round 1 Maximum: 1.00 at rounds 3, 4, 5, 6, 7, 8, 9, and 10 | Minimum: 0.6290 at round 1 Maximum: 1.00 at rounds 5, 7, 8, 9, and 10 | Minimum: 0.8233 at round 1 Maximum: 1.00 at rounds 6, 7, and 10 |
| **F1-score** | Minimum: 0.7488 at round 1 Maximum: 1.00 at round 10 | Minimum: 0.7363 at round 1 Maximum: 1.00 at rounds 8 and 10 | Minimum: 0.7368 at round 1 Maximum: 1.00 at round 9 | Minimum: 0.7593 at round 1 Maximum: 1.00 at round 10 | Minimum: 0.6857 at round 1 Maximum: 0.9898 at round 10 | Minimum: 0.7753 at round 1 Maximum: 1.00 at round 6 | Minimum: 0.6957 at round 1 Maximum: 0.995 at round 8 | Minimum: 0.7832 at round 1 Maximum: 1.00 at rounds 7 and 10 | Minimum: 0.6872 at round 1 Maximum: 0.9951 at round 8 | Minimum: 0.8369 at round 1 Maximum: 0.9974 at round 10 |
| **Loss** | Minimum: 0.0089 at round 10 Maximum: 0.5003 at round 1 | Minimum: 0.0058 at round 8 Maximum: 0.5097 at round 1 | Minimum: 0.0145 at round 10 Maximum: 0.6042 at round 1 | Minimum: 0.0090 at round 10 Maximum: 0.4989 at round 1 | Minimum: 0.0180 at round 8 Maximum: 0.6167 at round 1 | Minimum: 0.0102 at round 6 Maximum: 0.4207 at round 1 | Minimum: 0.0185 at round 8 Maximum: 0.4982 at round 1 | Minimum: 0.0121 at round 10 Maximum: 0.3963 at round 1 | Minimum: 0.0146 at round 10 Maximum: 0.6803 at round 1 | Minimum: 0.0094 at round 10 Maximum: 0.3241 at round 1 |

Whereas, the least valued of recall were resulted as 0.9886, 0.9901, 0.9902, and 0.995 by clients 5, 7, 9, and 10, respectively.

Fig 35 shows the precision of the model, where each client has shown a significant improvement in the achieved precision over rounds.

Fig 36 shows the round-wise F1-score of 10 different clients in FL-framework with non-IID data. The highest F1-score of 1.00 has been resulted by clients 1, 2, 3, 4, 6, and 8, whereas, clients 5, 7, 9, and 10 have resulted in highest F1-score value of 0.9898, 0.995, 0.9951, and 0.9974, respectively.

Fig 37 depicts the round-wise loss of 10 different clients in FL-framework with non-IID data. The highest loss of 0.5003, 0.5097, 0.6042, 0.4989, 0.6167, 0.4207, 0.4982, 0.3963, 0.6803, and 0.3241, whereas, the lowest loss of 0.0089, 0.0058, 0.0145, 0.0090, 0.0180, 0.0102, 0.0185, 0.0121, 0.0146, and 0.0094 have been resulted by clients 1, 2, 3, 4, 5, 6, 7, 8, 9, and 10, respectively.

## 4.5 Discussion and predictive analysis on the results identified by the proposed FL model

This section provides the outcomes of the proposed FL model. Table 8 tabulates the detailed summary on the results identified by the proposed FL-MobileNetV2 model implemented with 5 and 10 clients in IID and non-IID distributions.

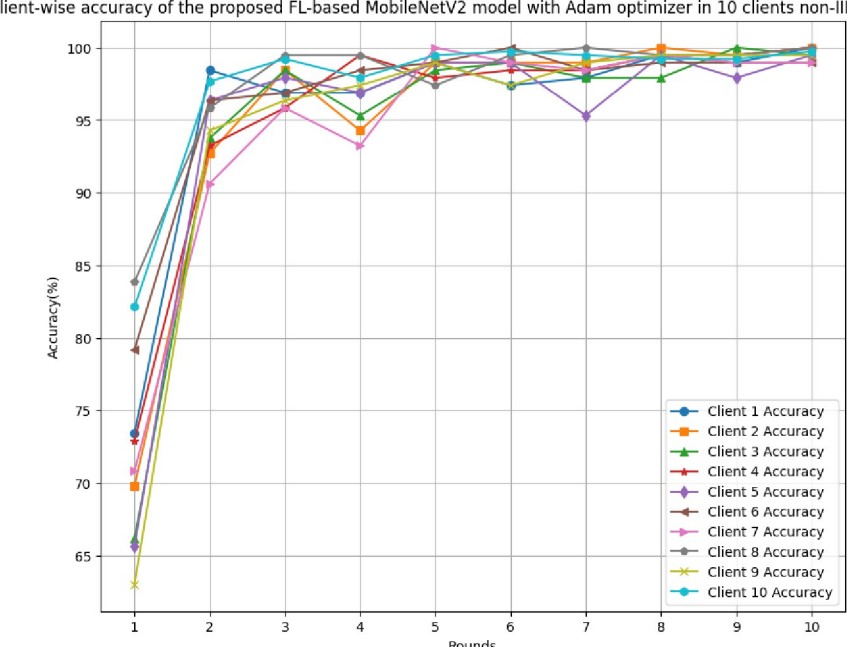

**Fig 33. Round-wise accuracy of 10 different clients in FL-framework with non-IID data.**

The outcomes of IID and non-IID distributions show that the accuracy achieved by IID distribution of the data among clients is outperforming in comparison to non-IID distribution of the data. This demonstrates that the model is optimal for IID distribution, however, the

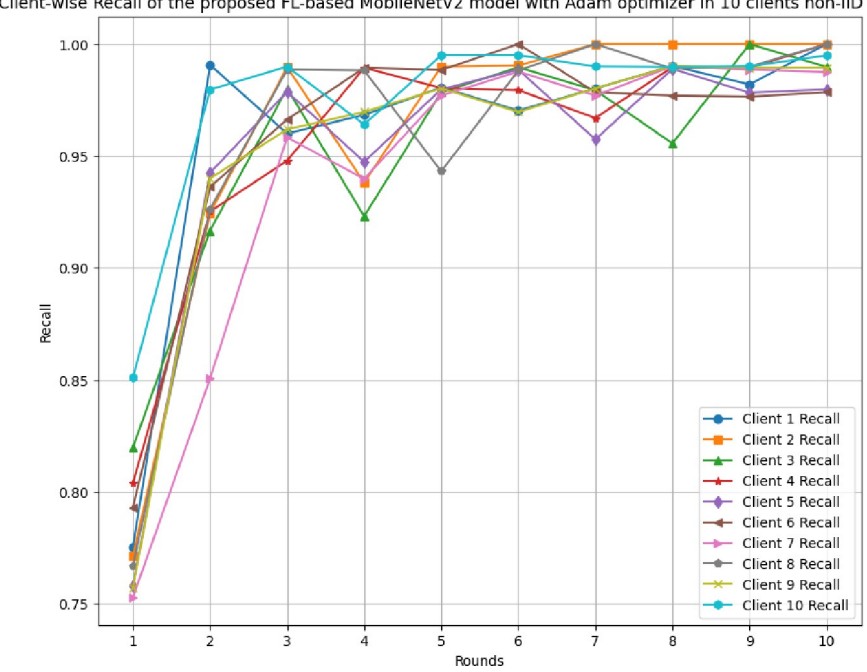

**Fig 34. Round-wise recall of 10 different clients in FL-framework with non-IID data.**

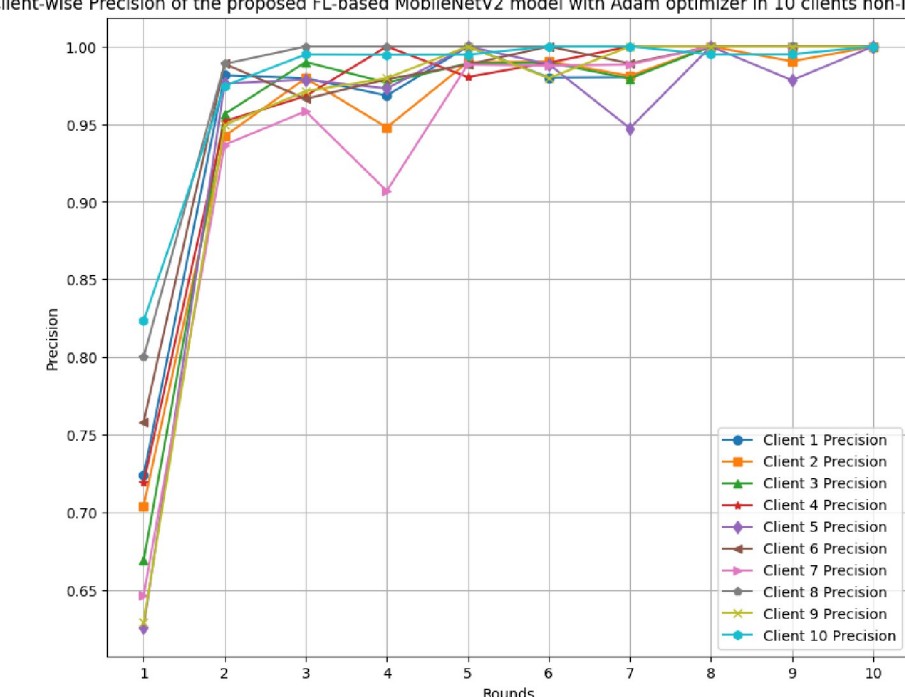

**Fig 35. Round-wise precision of 10 different clients in FL-framework with non-IID data.**

performance achieved on the non-IID distribution is also improved in comparison to the existing models. The non-IID is the most commonly used distribution in the real world scenarios, where, the clients participating in the architecture are from the different regions

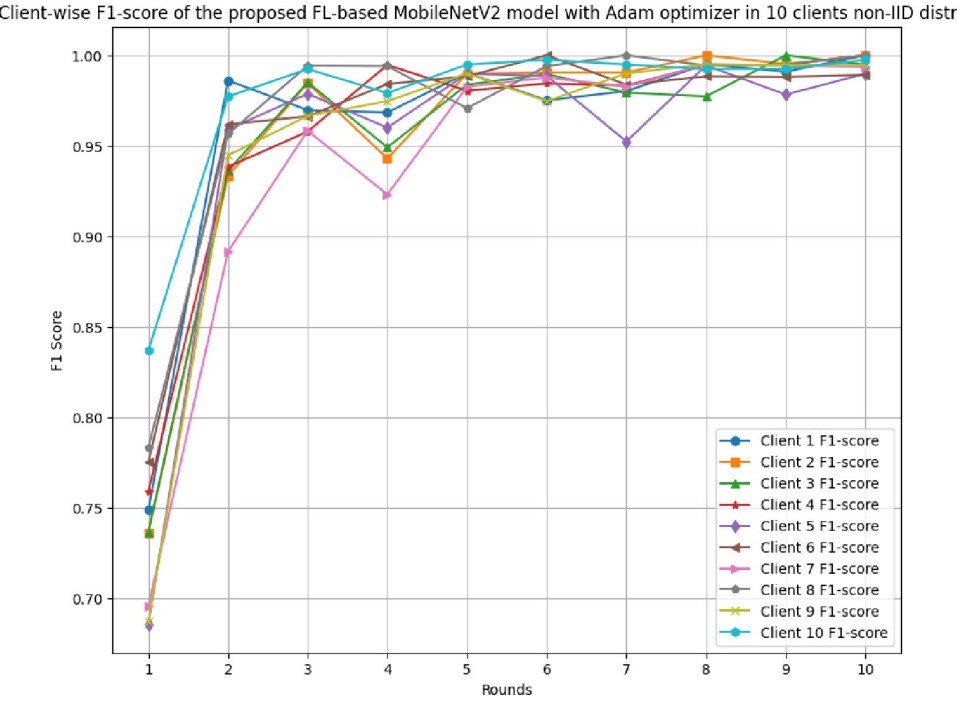

**Fig 36. Round-wise F1-score of 10 different clients in FL-framework with non-IID data.**

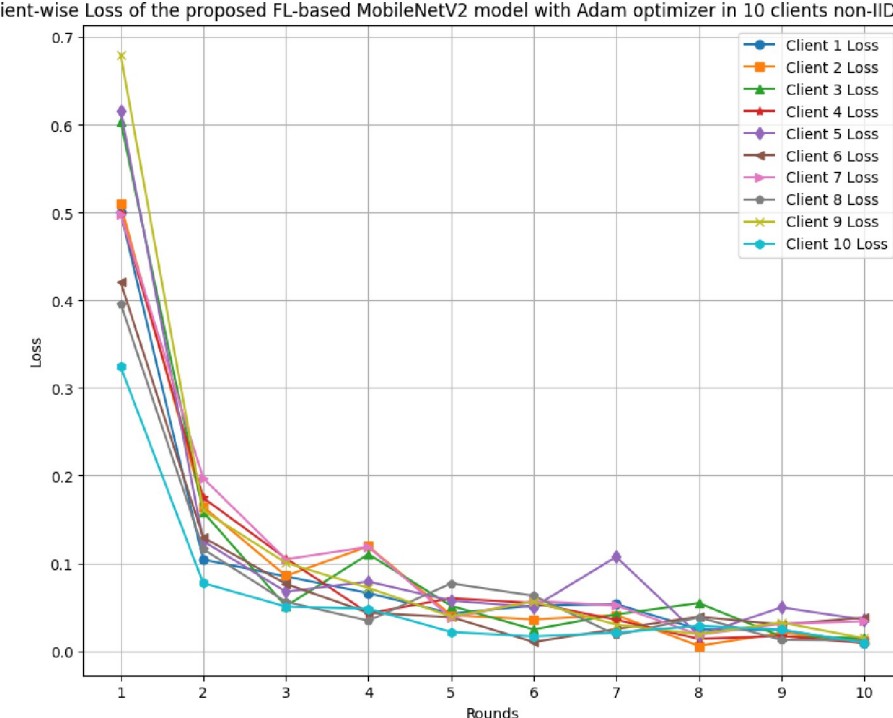

**Fig 37. Round-wise loss of 10 different clients in FL-framework with non-IID data.**

containing diverse datasets. It is one of the most important characteristics of the FL-architecture, however, handling non-IID types of data is challenging due to the demand of must generalizing the optimal outcomes on the skewed and diverse datasets from different clients.

The performance achieved by the proposed model on both IID and non-IID distributions are optimal, which shows that the model is highly efficient and adaptable for real world disease detection application in healthcare sectors. Though the model has resulted in the higher performance outcomes with the IID distribution, nevertheless, the data in the pragmatic scenarios are randomly distributed, which leads to the outcome that the proposed model with IID distribution is not compatible to be implemented in the realistic configurations as it follows the similar distribution of the data among clients. Conversely, the accuracy of non-IID data with 5 and 10 clients architecture have been identified as 99.71% and 99.61%, respectively, which depicts that the model is robust and optimal for handling real-world complexities and variations of the datasets.

In addition, the better generalization of the outcomes have been analyzed when less number of clients were used. In both distributions, the similar trend of the outcomes have been

**Table 8. A detailed summary on the results identified by the proposed FL-MobileNetV2 model.**

| | Accuracy | Recall | Precision | F1-score | Loss |
|---|---|---|---|---|---|
| **IID Data** | | | | | |
| **5 Clients** | 99.76% at round 9 | 0.997 at round 9 | 0.999 at round 8 | 0.9976 at round 9 | 0.0107 at round 9 |
| **10 Clients** | 99.64% at round 10 | 0.9954 at round 10 | 0.9973 at round 8 | 0.9959 at round 10 | 0.0139 at round 10 |
| **Non-IID Data** | | | | | |
| **5 Clients** | 99.71% at round 6 | 0.9962 at round 5 | 1.00 at round 6 | 0.998 at round 6 | 0.0107 at round 6 |
| **10 Clients** | 99.61% at round 10 | 0.9928 at round 8 | 0.9995 at round 8 | 0.9995 at round 10 | 0.0188 at round 10 |

identified. This happens because the increased number of clients add complexities, communication overhead, and scalability issues in the architecture. Increasing the number of clients also have resource constraints because the devices at the participating clients may range from lower capacity to high powered devices, which may result in difficulty in managing the FL-process across the varied types of devices. Additionally, increasing the client may also lead to slow model convergence leading to high time complexity and resource utilization. Hence, it is essential to optimally choose the client value for developing glioma brain tumour detection FL model.

## 4.6 Performance comparative analysis of the proposed FL-MobileNetV2 model with the existing FL-based tumour detection models

This section elaborates on the comparative analysis of the proposed FL-MobileNetV2 model with the existing FL-based tumour detection models.

Fig 38 depicts the comparative analysis of various FL-based tumour detection model. It has been observed that the proposed FL-MobileNetV2 model outperforms in both IID and non-IID distributions when compared with the existing model. The FL-centric CNN model with dataset 1 and dataset 2 provided in [27] shows the highest accuracy of 86% and 82%, respectively. The ensemble of InceptionV3, DenseNet, and VGG16 provided in [28] has also resulted in an accuracy of 91.05%, whereas, the model presented in [35] has shown an accuracy of 89% with the FL-VGG model. The proposed model implemented with 5 clients' architecture in IID

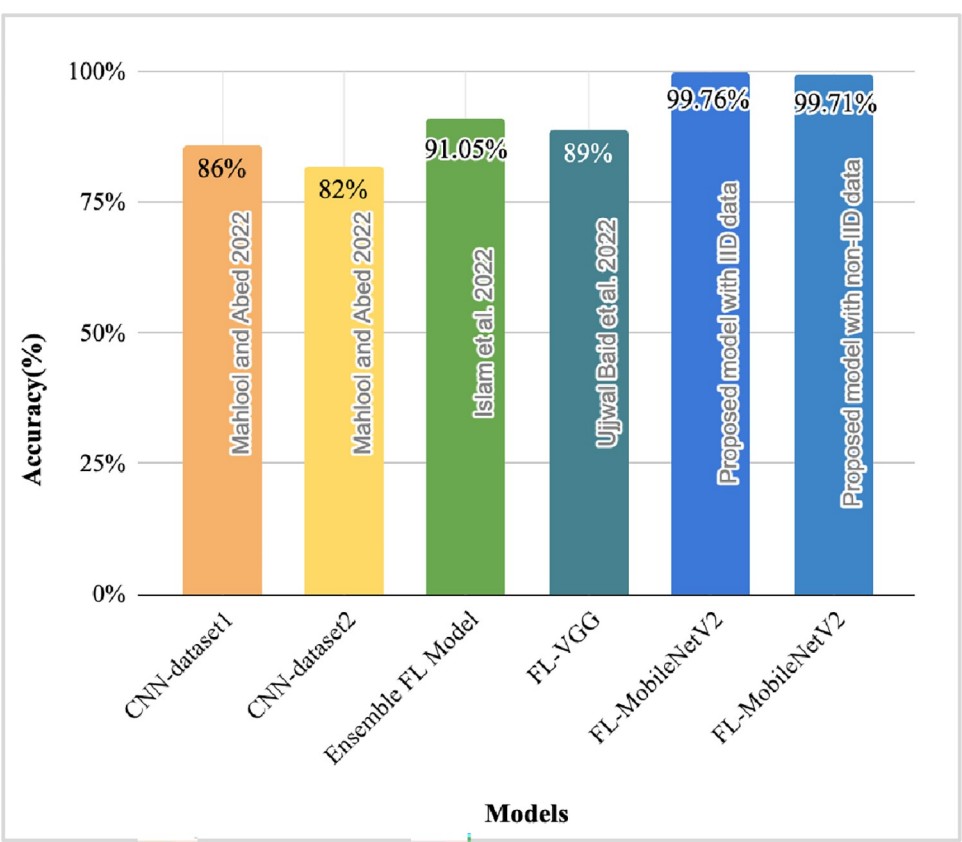

**Fig 38. Performance comparison of the proposed FL-MobileNetV2 model with the existing FL-based glioma brain tumour detection models.**

**Table 9. Ablation study of the proposed model with the existing glioma brain tumour detection models.**

| Ref. | Model | Client | Batch Size | Learning rate | Epoch | Rounds | Optimizer | Accuracy |
|---|---|---|---|---|---|---|---|---|
| [28] | Combined InceptionV3, DenseNet, VGG16 | 50 | 16 | 0.001 | 10 | 5 | Adam | 91.05% |
| [27] | CNN | 2 | 32 | 0.001 | 45 | NA | Adam | Dataset1: 86% Dataset2: 82% |
| [30] | FL-VGG | 8 | NA | 0.001 | 1 | 500 | Adam | Accuracy: 89% |
| Proposed FL-based Model with IID data | MobileNetV2 | 5 and 10 | 32 | 0.001 | 10 | 10 | Adam | Accuracy: 99.76% |
| Proposed FL-based Model with non-IID data | MobileNetV2 | 5 and 10 | 32 | 0.001 | 10 | 10 | Adam | Accuracy: 99.71% |

and non-IID distribution has shown an accuracy of 99.76% and 99.71%, which are better when compared with existing models. The comparative analysis results that the proposed model is robust and highly efficient in generalizing better and accurate outcomes for glioma brain tumour detection. Further, this model can be used in the healthcare sector with faster and more accurate training in diverse real-world data to improve privacy and reduce the critical concerns associated with the patient's sensitive information.

The models proposed in [27, 28, 30] have not been trained with the appropriate and balanced dataset, which directly impacts the model performance and results in convergence issues and overfitting to the majority class. Further, the small dataset has been utilized by the authors which may result in poor generalization if used for training a large number of clients in the architecture. The limited information on the hyperparameter tuning may constraints the model reproducibility. Therefore, the proposed model has been provided with the detailed hyperparameter configuration and pseudo code availability to reproduce and train this model with the large and diverse datasets. Further, the improved performance of the proposed model over the existing model shows that the proposed solution is better and results in optimal generalization of the outcomes.

Table 9 shows the ablation study of the proposed model with the state-of-the-art glioma brain tumour detection models. The proposed model outperforms the existing models, in terms of accuracy for both IID and non-IID distributions. The MobileNetV2 model is more efficient and lightweight in comparison to other existing models, where it contains the implementation of the FL-framework with two different client values as 5 and 10. Though the optimizer used by the existing models is also Adam, however, the base models utilized for training are highly complex which may outcome in higher computational overhead due to intricate architecture. Hence, the proposed model is capable of achieving improved accuracy in the distributed architecture by minimizing computational overhead in a lightweight base model FL framework.

## 5. Conclusion

As per the report published by the National Library of Medicine, it has been mentioned that in the years 1995–2018, a number of 31,922 deaths have been reported caused by glioma tumours. It originates in the central nervous system (CNS) impacting neurological functions, cognitive abilities, and causing seizures. Chemotherapy, radiation therapy, and immunotherapy are used for the treatment of glioma tumours, which are expensive and may cause allergic reactions due to ionizing radiation if done repetitively. In the proposed work, the privacy-protected and decentralized FL-based MobileNetV2 model has been employed for the early detection of malignant glioma tumours. This model was implemented with two different client architectures in IID and non-IID distributions. The hyperparameters configured in the

proposed model were Adam as the optimizer, a learning rate of 0.001, a batch size of 32, 10 local epochs, FedAVG for aggregation, 10 global epochs, and 10 rounds. This model has been identified as the most optimal performing model in both IID and non-IID distributions with 5 and 10-clients horizontal FL architecture. In the 5-clients architecture, the model has shown an accuracy of 99.76% and 99.71%, whereas, for the 10-clients, it has been achieved as 99.64% and 99.61% for IID and non-IID distributions, respectively. These results affirm that the proposed model outperforms the existing FL-based tumour detection models by generalizing improved outcomes with the FedAVG aggregation technique. In the future, the proposed model may be implemented using vertical federated learning and transfer federated learning models for handling the much higher level of heterogeneity for varied feature spaces, respectively.

## Author Contributions

**Conceptualization:** Shagun Sharma.

**Data curation:** Shagun Sharma.

**Funding acquisition:** Ali Nauman.

**Investigation:** Shagun Sharma, Kalpna Guleria, Ali Nauman.

**Methodology:** Shagun Sharma, Kalpna Guleria, Sapna Juneja, Swati Kumari, Ali Nauman.

**Project administration:** Ayush Dogra.

**Resources:** Ayush Dogra, Swati Kumari.

**Software:** Swati Kumari.

**Supervision:** Deepali Gupta, Sapna Juneja.

**Validation:** Deepali Gupta, Sapna Juneja.

**Visualization:** Deepali Gupta.

**Writing – original draft:** Sapna Juneja, Swati Kumari, Ali Nauman.

**Writing – review & editing:** Deepali Gupta, Sapna Juneja, Swati Kumari, Ali Nauman.

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
