## [Decision Letter · Decision Letter 0]

13 Aug 2024

PONE-D-24-28560A Privacy-preserved Horizontal Federated Learning for Malignant Glioma Tumour Detection using Distributed Data-silosPLOS ONE

Dear Dr. kumari,

Thank you for submitting your manuscript to PLOS ONE. After careful consideration, we feel that it has merit but does not fully meet PLOS ONE’s publication criteria as it currently stands. Therefore, we invite you to submit a revised version of the manuscript that addresses the points raised during the review process.

We look forward to receiving your revised manuscript.

Kind regards,

Subramani Neelakandan

Academic Editor

PLOS ONE

Journal Requirements:

**Additional Editor Comments:**

Author presented the research work "Privacy-preserved Horizontal Federated Learning for Malignant Glioma Tumour Detection using Distributed Data-silos". I have found some major flaws in the manuscript especially many typo errors and others. Every acronym should be defined the first time it appears for example FL-based, FL should abbreviated first time it appears similarly other abbreviations too. Author should discuss the recent pubished works in the introduction section. In the table.1, author just used symbol for preprocessing and data balancing instead of describing the process like how they done the pre0processing ? whether author used any segmentation or filters or labeling etc. In section 3.1.1 Author discussed about 224X224X3. is the right way to describe the image size ? Similarly the statement "The collected image and pre-processed image sample have been shown in Fig. 1." Author discussing about single image or general process of pre-processing images ?. Many statements are not conveying the exact meaning. Author need to describe properly about privacy-preserved in Fig.2. How this privacy-preserved was executed in Fig.4 too?

Also author stated that "the weights from each client are collected and aggregated using the FedAVG technique and passed to the server for global model update". Here what is the initial weight value assigned for each client and what is the base for this ? In section 4.2, all the mathematical formula was dis appeared and showing poor presentation. Author must verify the pdf file or original file before submitting to the journal. In table.3 the performance paramete accuracy alone represented in % where as other parameters presented in normal value , what is the reason ? Also result presented for only 5 clients and what about remaining clients ? Also the values are getting confused to identify the exact performance of each clients, for example C1 achieved the accuracy of 92.79% to 99.76%. Here what is the actual predicted value ? you mean that the predicted range may start from 92.79% to 99.76%. So as a reviewer how we can understand the value claimed by C1? Whether table values presented round wise or epoch? if the values are round wise then why not to represent round basis ?

Author should describe the detailed experimental setup information for this result and processing. All the graph showing that after 2nd round all the clients achieved the accuracy 90% what is the real scenario behind this ? Author applied any special features for this or any model evaluation techniques applied ?

Author may include the model evaluation or fitness score for this research. Fig.15,16,17 and 18 can be drawn as single image so that the length of the manuscript may be reduced similarly other images too. Table.4 too discuss the same scenarionof table.2.

Overall author failed to discuss most important discussion about the privacy preserved result parameters. Instead of this author described too many result which is not required to prove the novelty of the research. Its better to discuss the ablation study and optimization of the proposed model with the state of the art instead of too many result with the different rounds and different clients.

Reviewers' comments:

Reviewer's Responses to Questions

**Comments to the Author**

1. Is the manuscript technically sound, and do the data support the conclusions?

Reviewer #1: Yes

2. Has the statistical analysis been performed appropriately and rigorously? 

Reviewer #1: Yes

3. Have the authors made all data underlying the findings in their manuscript fully available?

Reviewer #1: Yes

4. Is the manuscript presented in an intelligible fashion and written in standard English?

Reviewer #1: Yes

5. Review Comments to the Author

**Reviewer #1:** 1- You need to explain the structure of the proposed system with simplified steps in the abstract

2- Ensure that the scientific term and its abbreviation are included only when they first appear

3-Point 3 listed below is not considered a contribution, but rather results reached by the researcher:

The outcomes of the proposed FL model have been analyzed in terms of accuracy, recall, loss, precision, and F1-score. The predictive study of the two data distribution configurations identified that in both IID and non-IID distributions, the better generalization of the accuracy was analysed with 5-client architecture as 99.76% and 99.71%, whereas, for 10 clients architecture, it has been achieved as 99.64% and 99.61%, respectively.

4- Ensure that the research in the section Literature review on tumor detection federated learning frameworks is arranged according to the year of publication (from oldest to newest), in addition to listing the weak points of this research during its review.

5- Insert a detailed table to divide the Dataset.

6- Verify the symbols for the equations mentioned in Results and Discussion section

7- Better clarify future work

6. PLOS authors have the option to publish the peer review history of their article (what does this mean?). If published, this will include your full peer review and any attached files.

Reviewer #1: No

---

## [Author Response · Author response to Decision Letter 0]

30 Sep 2024

Editor’s Comments

1. Author presented the research work "Privacy-preserved Horizontal Federated Learning for Malignant Glioma Tumour Detection using Distributed Data-silos". I have found some major flaws in the manuscript especially many typo errors and others. Every acronym should be defined the first time it appears for example FL-based, FL should abbreviated first time it appears similarly other abbreviations too.

Response:

Done!

Dear Editor, Thank you for your suggestion. 

As per the suggestion, the manuscript has been proofread for typo errors.

Further, all acronyms are used at their first use, and this change has been applied consistently throughout the manuscript.

2. Author should discuss the recent pubished works in the introduction section.

Response:

Done!

Dear Editor, 

Thank you for your suggestion. 

In accordance with the suggestion, recent references relevant to the subject area have been cited in the introduction section as references [7], [9], [22], [25], and [26] on pages 3 and 4, as highlighted in green.

3. In the table.1, author just used symbol for preprocessing and data balancing instead of describing the process like how they done the pre0processing ? whether author used any segmentation or filters or labeling etc.

Response:

Done!

Dear Editor, 

Thank you for the suggestion! 

In accordance with the suggestion, the pre-processing and data-balancing techniques used in the existing works have been included in Table 1 on pages 8-15, as highlighted in green.

4. In section 3.1.1 Author discussed about 224X224X3. is the right way to describe the image size ?

Response:

Done!

Dear Editor, 

Thank you for the suggestion!

The grayscale images were used to train the proposed model. These images are resized to 224X224X1; however, the size was mistakenly written as 224X224X3 in the manuscript. This error has now been corrected and updated to 224X224X1 on pages 4, 16, and 23.

5. Similarly the statement "The collected image and pre-processed image sample have been shown in Fig. 1." Author discussing about single image or general process of pre-processing images ?.

Response:

Dear Editor, 

The statement “The collected image and pre-processed image sample have been shown in Fig. 1” is intended to illustrate examples of the original and pre-processed images from the entire dataset. This represents the general pre-processing technique that has been applied to the complete dataset.

6. Many statements are not conveying the exact meaning. Author need to describe properly about privacy-preserved in Fig.2. How this privacy-preserved was executed in Fig.4 too?

Response:

Done!

Dear Editor, 

Thank you for the suggestion!

In accordance with the suggestion, the detailed description of privacy preservation in Fig. 2 and Fig. 4 has been added on pages 18 and 23, as highlighted in green.

Healthcare providers follow data privacy preservation acts, including HIPAA in the United States, the General Data Protection Regulation (GDPR) in the European Union, the Personal Information Protection and Electronic Documents Act (PIPEDA) in Canada, and the Digital Personal Data Protection Act. (DPDT 2023) in India. The proposed federated framework also follows privacy preservation making it technically suitable for disease prediction, without sharing the data to the server or any other peer clients.

The proposed model preserves privacy in such a way that this framework contains numerous hospitals and a central server where each hospital contains its own private dataset at a personalized location. The data contained within each client is neither shared with the central server nor with the other participating clients. The federated process starts from the central server containing the base model, which is distributed to each participating clients to train using a personalized dataset. The clients use their private dataset for training the base model and share the model updates with the central server, which then performs the updates aggregation and updates the global model with the aggregated data. Further, this process continues until the federated framework achieves the most optimal accuracy for glioma detection in a privacy-preserving architecture. This framework shows the model training process in a decentralized manner, where, the client’s data is private and neither shared with the central server nor with other clients.

7. Also author stated that "the weights from each client are collected and aggregated using the FedAVG technique and passed to the server for global model update". Here what is the initial weight value assigned for each client and what is the base for this ?

Response:

Done!

Dear Editor, 

Thank you for the suggestion!

In accordance with the comment, the description regarding the base of the initial weights has been addressed in section 3.2 on pages 17 and 18, as highlighted in green color.

8. In section 4.2, all the mathematical formula was disappeared and showing poor presentation. Author must verify the pdf file or original file before submitting to the journal.

Response:

Done!

Dear Editor,

Thank you for your valuable suggestion!

In accordance with the suggestion, the equations have been updated throughout the manuscript on pages 21, 22, and 26, as highlighted in green. 

9. In table.3 the performance paramete accuracy alone represented in % where as other parameters presented in normal value , what is the reason ?

Response:

Dear Editor, 

Thank you for your valuable analysis!

In Tables 3 and 4 (now updated to Table. 4 and Table. 5), accuracy is represented as a percentage, while other parameters are shown as decimal values. 

In the proposed work, the accuracy parameter is expressed as a percentage to indicate the proportion of correct predictions, reflecting the model’s performance. It is a standard convention in the area of deep learning to depict accuracy as a percentage.

In contrast, Precision, Recall, and F1-Score are presented as decimal values to align with their inherent scale and calculation methods, ensuring consistency.

Therefore, the accuracy of the proposed model has been depicted as a percentage and other parameters as decimal values. 

10. Also result presented for only 5 clients and what about remaining clients ?

Response:

 Done!

Dear Editor, 

Thank you for your valuable analysis!

The proposed FL-based MobileNetV2 model was trained using both 5-client and 10-client architectures. Results for the 5-client setup are presented in Tables 4 and 6, while results for the 10-client framework are shown in Tables 5 and 7, located on pages 29, 36, 37, 44, 45, 51, 52, and 53, as highlighted in green. Additionally, performance graphs for the 5-client and 10-client architectures are provided in sections 4.1 and 4.2 for IID data and in sections 4.3 and 4.4 for non-IID data.

11. Also the values are getting confused to identify the exact performance of each clients, for example C1 achieved the accuracy of 92.79% to 99.76%. Here what is the actual predicted value ? you mean that the predicted range may start from 92.79% to 99.76%. So as a reviewer how we can understand the value claimed by C1? Whether table values presented round wise or epoch? if the values are round wise then why not to represent round basis?

Response:

Done!

Dear Editor, 

Thank you for your valuable suggestion!

As per the suggestion, the accuracy as well as other performance parameters, have been updated in the round-wise in Tables 4 on pages 28 and 29, Table 5 on pages 35, 36, and 37, Table 6 on pages 44 and 45, and Table 7 on pages 51, 52, and 53, as highlighted in green.

12. Author should describe the detailed experimental setup information for this result and processing.

Response:

Done!

Dear Editor, 

Thank you for your valuable suggestion!

In accordance with the suggestion, the experimental setup information for results and processing has been included in Section 3.2 on page 19, highlighted in green. Additionally, a detailed hyperparameter configuration is provided in Table 3 on page 25, also highlighted in green.

13. All the graph showing that after 2nd round all the clients achieved the accuracy 90% what is the real scenario behind this ? Author applied any special features for this or any model evaluation techniques applied ?

Response:

Dear Editor, 

Thank you for your valuable comment!

Yes, in the proposed federated learning framework, we have implemented an optimized version of the FedAvg algorithm with the Adam adaptive learning rate to accelerate convergence during the model's development. Further, the base MobileNetV2 model is also initialized with pre-trained weights from the ImageNet dataset, which contributes to achieving over 90% accuracy after the second round.

14. Author may include the model evaluation or fitness score for this research. Fig.15,16,17 and 18 can be drawn as single image so that the length of the manuscript may be reduced similarly other images too.

Response:

Done!

Dear Editor, 

Thank you for your valuable suggestion!

The fitness score for the proposed model has been calculated based on accuracy, precision, recall, loss, and F1-score, as detailed in Section 4 ("Results and Discussion"). This section includes fitness scores for each client and for the entire FL framework, covering both IID and non-IID data distributions across 5-client and 10-client architectures.

In response to the suggestion, the graphs have been consolidated into single images. Specifically, accuracy has been represented separately in Fig. 5, Fig. 15, Fig. 25, and Fig. 35 (now updated to Fig. 5,

● Fig. 13, Fig. 21, and Fig. 29), which is shown as a percentage. Additionally:

● Fig. 6, 7, and 8 have been combined into a single image and replaced with Fig. 6 on page 27.

● Fig. 16, 17, and 18 have been merged into a single image and replaced with Fig. 14 on page 34.

● Fig. 26, 27, and 28 have been combined into one image and replaced with Fig. 22 on page 43.

● Fig. 36, 37, and 38 have been merged into one image and replaced with Fig. 30 on page 50.

15. Table.4 too discuss the same scenarionof table.2.

Response:

Done!

Dear Editor, 

Thank you for sparing your valuable time in reviewing the manuscript.

We would like to clarify that Table 2 (now updated to Table. 3) and Table 4(now updated to Table. 5) serve different purposes: Table 2(now updated to Table. 3) details the hyperparameter configurations for developing the proposed federated learning-based glioma brain tumor detection framework, while Table 4(now updated to Table. 5) presents a comprehensive analysis of the round-wise performance outcomes for the 10-client federated framework with IID data distribution.

16. Overall author failed to discuss most important discussion about the privacy preserved result parameters. Instead of this author described too many result which is not required to prove the novelty of the research.

Response:

Dear Editor, 

Thank you for the suggestion!

In the context of federated architecture,

Data privacy refers to the protection of personal or sensitive information from unauthorized access or misuse, ensuring that individuals' data is handled in compliance with privacy laws and regulations, whereas, 

In the proposed work, federated learning has been used for data privacy, where the data at each hospital is neither shared with the server nor with peer clients. Only the updates have been shared with the central server instead of the actual data. This shows that the proposed model follows strict data privacy preservation acts, including HIPAA in the United States, the General Data Protection Regulation (GDPR) in the European Union, the Personal Information Protection and Electronic Documents Act (PIPEDA) in Canada, and the Digital Personal Data Protection Act. (DPDT 2023) in India. 

As per the suggestion, the discussion about the proposed privacy-preserved federated learning architecture has been presented in section 3 on pages 18 and 23, as highlighted in green.

17. Its better to discuss the ablation study and optimization of the proposed model with the state of the art instead of too many result with the different rounds and different clients.

Response:

Done!

Dear Editor, 

Thank you for your valuable suggestion!

In response to your valuable suggestion, the ablation study and optimization of the proposed model compared to state-of-the-art models have been included in Section 4.6 on pages 60 and 61, as highlighted in green.

Additionally, to reduce the number of images in the manuscript, several figures have been combined.

The updated figure details are as follows:

● Fig. 6, 7, and 8 have been combined into a single image and replaced with Fig. 6 on page 27.

● Fig. 16, 17, and 18 have been merged into a single image and replaced with Fig. 14 on page 34.

● Fig. 26, 27, and 28 have been combined into one image and replaced with Fig. 22 on page 43.

● Fig. 36, 37, and 38 have been merged into one image and replaced with Fig. 30 on page 50.

#Reviewer’s Comments

1. You need to explain the structure of the proposed system with simplified steps in the abstract

Response:

Done!

Dear reviewer, 

Thank you for your suggestion.

Following the suggestion, the comment has been addressed in the abstract on pages 1 and 2, as highlighted in green color.

2. Ensure that the scientific term and its abbreviation are included only when they first appear

Response:

Done!

Dear reviewer, Thank you!

In accordance with the suggestion, scientific terms have been abbreviated before their first use throughout the manuscript.

3. Point 3 listed below is not considered a contribution, but rather results reached by the researcher: The outcomes of the proposed FL model have been analyzed in terms of accuracy, recall, loss, precision, and F1-score. The predictive study of the two data distribution configurations identified that in both IID and non-IID distributions, the better generalization of the accuracy was analysed with 5-client architecture as 99.76% and 99.71%, whereas, for 10 clients architecture, it has been achieved as 99.64% and 99.61%, respectively.

Response:

Done!

Dear Reviewer, 

Thank you for your valuable suggestion!

In response to the suggestion, the contribution of the proposed work has been updated in section 1 on page 4, as highlighted in green color.

4. Ensure that the research in the section Literature review on tumor detection federated learning frameworks is arranged according to the year of publication (from oldest to newest), in addition to listing the weak points of this research during its review.

Response:

Done!

Dear Reviewer, 

Thank you for the valuable suggestion.

As suggested, the changes have been incorporated into Section 2 on pages 6-15. Additionally, the weaknesses and research gaps in existing models have been detailed in Section 2, following Table 1 on page 15, as highlighted in green. These research gaps are also summarized in the last column, "Future Scope," of Table 1.

5. Insert a detailed table to divide the Dataset.

Response:

Done!

Dear Reviewer, 

Thank you for the valuable suggestion.

In accordance with the suggestion, a detailed description of the dataset distribution has been added to Table. 2 in section 3.1 on page 16 as highlighted in green color. 

6. Verify the symbols for the equations mentioned in Results and Discussion section

Response:

Done!

Dear reviewer, 

Thank you for the suggestion!

In accordance with the suggestion, the symbols and equations presented in the results and discussion section have been verified and updated in section 4 on page 26, as highlighted in green color.

7. Better clarify future work

Response:

Done!

Dear reviewer, 

Thank you for the suggestion!

In accordance with the suggestion, future work has been added to the conclusion section on page 62, as highlighted in green.

Submitted for your kind information and action.

We will be highly obliged if we are provided with the opportunity to publish our research work through your esteemed journal.

---

## [Decision Letter · Decision Letter 1]

29 Oct 2024

PONE-D-24-28560R1A Privacy-preserved Horizontal Federated Learning for Malignant Glioma Tumour Detection using Distributed Data-silosPLOS ONE

Dear Dr. kumari,

Thank you for submitting your manuscript to PLOS ONE. After careful consideration, we feel that it has merit but does not fully meet PLOS ONE’s publication criteria as it currently stands. Therefore, we invite you to submit a revised version of the manuscript that addresses the points raised during the review process.

We look forward to receiving your revised manuscript.

Kind regards,

Subramani Neelakandan

Academic Editor

PLOS ONE

Journal Requirements:

Additional Editor Comments:

Many of my comments not addressed properly with the required information so please address the below comments carefully with the valid information without fail.

In the table.1, author just used symbol for preprocessing and data balancing instead of describing the process like how they done the preprocessing ? whether author used any segmentation or filters or labeling etc. In section 3.1.1 Author discussed about 224X224X3. is the right way to describe the image size ? Similarly the statement "The collected image and pre-processed image sample have been shown in Fig. 1." Author discussing about single image or general process of pre-processing images ?. Many statements are not conveying the exact meaning. Author need to describe properly about privacy-preserved in Fig.2. How this privacy-preserved was executed in Fig.4 too?

Also author stated that "the weights from each client are collected and aggregated using the FedAVG technique and passed to the server for global model update". Here what is the initial weight value assigned for each client and what is the base for this ?

Reviewers' comments:

Reviewer's Responses to Questions

**Comments to the Author**

1. If the authors have adequately addressed your comments raised in a previous round of review and you feel that this manuscript is now acceptable for publication, you may indicate that here to bypass the “Comments to the Author” section, enter your conflict of interest statement in the “Confidential to Editor” section, and submit your "Accept" recommendation.

Reviewer #1: All comments have been addressed

2. Is the manuscript technically sound, and do the data support the conclusions?

Reviewer #1: Yes

3. Has the statistical analysis been performed appropriately and rigorously? 

Reviewer #1: Yes

4. Have the authors made all data underlying the findings in their manuscript fully available?

Reviewer #1: Yes

5. Is the manuscript presented in an intelligible fashion and written in standard English?

Reviewer #1: Yes

6. Review Comments to the Author

Reviewer #1: (No Response)

7. PLOS authors have the option to publish the peer review history of their article (what does this mean?). If published, this will include your full peer review and any attached files.

Reviewer #1: No

---

## [Author Response · Author response to Decision Letter 1]

19 Nov 2024

Comments Response

In the table.1, author just used symbol for preprocessing and data balancing instead of describing the process like how they done the pre0processing ? whether author used any segmentation or filters or labeling etc. Done!

Dear Editor, 

Thank you for the valuable suggestion! 

In accordance with your suggestion, we have thoroughly reviewed the existing literature and incorporated the relevant pre-processing methods and data-balancing techniques utilized in the existing works at the end of section 2.2 before Table 1. The same has been highlighted in green color on pages 8-16. 

In section 3.1.1 Author discussed about 224X224X3. is the right way to describe the image size ? Done!

Dear Editor, 

Thank you for the suggestion!

As per the suggestion provided in the previous revision, we have updated the image dimensions to 224X224X1, which is intended to show the dimensions of an image tensor used as input to the EfficientNetB3 model.

However, in section 3.1.1, we want to show the spatial dimensions of the image, hence, it has been updated to 224X224 pixels for height and width, respectively, on pages 4, 18, and 25, as highlighted in green color.

Similarly the statement "The collected image and pre-processed image sample have been shown in Fig. 1." Author discussing about single image or general process of pre-processing images ?. Done!

Dear Editor, 

The statement “The collected image and pre-processed image sample have been shown in Fig. 1” is intended to illustrate examples of the original and pre-processed images from the entire dataset. However, for simplification and better understanding, the statement has been updated to “An example of the initially collected images and the resized images has been illustrated in Fig. 1. The dataset has 2 classes, where the example of resizing the images have been shown as; sample 1 and sample 2 represent the resizing of actual no-tumour image to 224X224 pixels and sample 3 and sample 4 shows the resizing of actual glioma image to 224X224 pixels.”

This represents the general pre-processing technique that has been applied to the complete dataset.

The same has been updated in section 3.1.1 and in Fig 1 as highlighted in green on page 18. 

Many statements are not conveying the exact meaning. Author need to describe properly about privacy-preserved in Fig.2. How this privacy-preserved was executed in Fig.4 too? Done!

Dear Editor, 

Thank you for the suggestion!

In accordance with the suggestion, the detailed description of privacy preservation in Fig. 2 and Fig. 4 has been added on pages 20 and 25, as highlighted in green. Further, an additional Figure, “Fig 5 Flow diagram of the proposed FL-based glioma detection model” has also been added in section 3.2.2 for a better understanding of the proposed privacy-preserved federated learning architecture, as highlighted in green on pages 27 and 28.

The proposed model preserves privacy in such a way that this framework contains numerous hospitals and a central server where each hospital contains its own private dataset at a personalized location. The data contained within each client is neither shared with the central server nor with the other participating clients. The federated process starts from the central server containing the base model, which is distributed to each participating client to train using a personalized dataset. The clients use their private dataset to train the base model and share the model updates with the central server instead of sharing the actual data. The central server uses a FedAVG aggregation for update aggregation. This aggregated value is used to update the global model. Further, this process continues until the federated framework achieves the most optimal accuracy for glioma detection in a privacy-preserving architecture. This framework shows the model training process in a decentralized manner, where the client’s data is private and is not shared with the central server or with other clients. 

Also author stated that "the weights from each client are collected and aggregated using the FedAVG technique and passed to the server for global model update". Here what is the initial weight value assigned for each client and what is the base for this ? Done!

Dear Editor, 

Thank you for the suggestion!

We have updated the manuscript to specify that each client in the federated architecture is provided with the initial global model’s weights, which were trained on the ImageNet dataset on the server. 

Additionally, as per your suggestion, we have revised the description regarding the initial weight values in Section 3.2 on page 19 and highlighted these updates in green color. The same has been mentioned below for your kind information:

The proposed work introduces a FL-based MobileNetV2 model for brain tumour glioma classification from non-tumour MRI scans. The base MobileNetV2 model has been initially trained with the ImageNet dataset on the central server and distributed to the participating client in the FL framework. Each layer in the MobileNetV2 has been initialized with the pre-trained weights of the ImageNet dataset, which contains 1 million images and 1000 different classes. The MobileNetV2 model training using the ImageNet dataset is responsible for configuring the initial weights to the server and participating clients. The range of these weights is between min = -1.061939001083374 to max = 1.4804697036743164. These weights obtained from the ImageNet dataset provide better model training and faster convergence. In the proposed FL-based model, the initial weight values assigned to each client remain the same as in the global model. These weight values to each client typically come from the central server, where global model initialization has been performed using a pre-trained MobileNetV2 model trained with the ImageNet dataset. In the proposed FL-model, the initial weights distributed to each client are the same as the global MobileNetV2 model trained using ImageNet dataset. It represents that each client receives and starts training with the same weights of the initial global model developed on the server side. These weight values provide the model with information regarding the classification task and lead to optimal outcomes as the model is already trained using a large dataset.

---

## [Editor Report · Decision Letter 2]

13 Dec 2024

A Privacy-preserved Horizontal Federated Learning for Malignant Glioma Tumour Detection using Distributed Data-silos

PONE-D-24-28560R2

Dear Dr. kumari,

We’re pleased to inform you that your manuscript has been judged scientifically suitable for publication and will be formally accepted for publication once it meets all outstanding technical requirements.

Kind regards,

Subramani Neelakandan

Academic Editor

PLOS ONE
---

## [Editor Report · Acceptance letter]

10 Jan 2025

PONE-D-24-28560R2 

PLOS ONE

Dear Dr. kumari, 

I'm pleased to inform you that your manuscript has been deemed suitable for publication in PLOS ONE. Congratulations! Your manuscript is now being handed over to our production team.

Kind regards, 

on behalf of

Dr. Subramani Neelakandan 

Academic Editor

PLOS ONE